# Analyzing cloud base at local and regional scales to understand tropical montane cloud forest vulnerability to climate change

Ashley E. Van Beusekom[1], Grizelle González[1], Martha A. Scholl[2]

[1]USDA Forest Service International Institute of Tropical Forestry, Jardín Botánico Sur, 1201 Calle Ceiba, Río Piedras, Puerto Rico 00926, United States
[2]US Geological Survey National Research Program, 12201 Sunrise Valley Drive, Reston, VA 20192, United States

*Correspondence to*: Ashley Van Beusekom (ashleyvanbeusekom@fs.fed.us)

**Abstract.** The degree to which cloud immersion provides water in addition to rainfall, suppresses transpiration, and sustains tropical montane cloud forests (TMCFs) during rainless periods is not well understood. Climate and land use changes represent a threat to these forests if cloud base altitude rises as a result of regional warming or deforestation. To establish a baseline for quantifying future changes in cloud base, we installed a ceilometer at 100 m altitude in the forest upwind of the TMCF that occupies an altitude range from ~600 m to the peaks at 1100 m in the Luquillo Mountains of Eastern Puerto Rico. Airport ASOS ceilometer data, radiosonde data, and CALIPSO satellite data were obtained to investigate seasonal cloud base dynamics, altitude of the trade-wind inversion and typical cloud thickness for the surrounding Caribbean region. Cloud base is rarely quantified near mountains, so these results represent a first look at seasonal and diurnal cloud base dynamics for the TMCF. From May 2013 – August 2016, cloud base was lowest during the mid-summer dry season, and cloud bases were lower than the mountaintops as often in the winter dry season as in the wet seasons. The Luquillo forest low cloud base altitudes were higher than six other sites in the Caribbean by ~200-600 m, highlighting the importance of site selection to measure topographic influence on cloud height. Proximity to the oceanic cloud system where shallow cumulus clouds are seasonally invariant in altitude and cover; along with local trade-wind orographic lifting and cloud formation, may explain the dry season low clouds. The results indicate climate change threats to low-elevation TMCFs are not limited to the dry season; changes in synoptic-scale weather patterns that increase frequency of drought periods during the wet seasons (periods of higher cloud base) may also impact ecosystem health.

## 1 Introduction

Mountains play a key role in collecting atmospheric moisture in tropical regions (Wohl et al., 2012). Around 500 TMCFs have been identified world-wide on mountains with frequent cloud cover; these can be at higher elevation (on larger mountains) which have lower temperatures, or lower elevation (on smaller mountains) which have more rainfall (Jarvis and Mulligan, 2011). The global set of TMCFs are almost all within 350 km of a coast and topographically exposed to higher-humidity air (Jarvis and Mulligan, 2011). Smaller mountains are likely to have clouds at lower elevations due to slightly higher adiabatic lapse rates (more temperature loss with the same elevation gain) than larger mountains, which undergo

greater heating of the land mass (the mass-elevation effect: Foster, 2001; Jarvis and Mulligan, 2011). This effect and the higher humidity near the ocean support TMCFs on small coastal mountains..

Up to 60% of the moisture input to TMCFs is derived from cloud water interception from low clouds (Bruijnzeel et al., 2011), but this value varies greatly from forest to forest and within an individual forest. Cloud water has been deemed critical for the health of TMCFs, specifically for the abundant epiphytes which require consistent moisture input from the atmosphere (Gotsch et al., 2015). Cloud water interception (including fog) adds moisture directly to the soil through the process of canopy interception and fog drip (Giambelluca et al., 2011), and indirectly alters the moisture budget through foliar uptake (Eller et al., 2013), lowering the saturation deficit of the atmosphere, and suppressing transpiration (Alvarado-Barrientos et al., 2014). For the low-elevation TMCF of this study, it might be expected that, given a relatively constant seasonal temperature lapse rate (within 0.1°C/1000 m for each season using previous study data (Van Beusekom et al., 2015)), wet seasons would have higher relative humidity caused by the larger amounts and spatial extent of rain events, and therefore lower clouds. Yet, previous field-based studies at four different low-elevation TMCFs found that similar or higher absolute amounts of fog precipitation were deposited during dry season measurements than during wet season measurements (Cavelier and Goldstein, 1989).

With their large energy inputs and fast rates of spatial and temporal change, tropical regions are some of the most dynamic atmospheric and hydrologic systems on Earth and thus may be significantly affected by climate change (Perez et al., 2016; Wohl et al., 2012). Regional cloud mass has been projected to decline with continued deforestation in maritime climates (van der Molen et al., 2006), and warming temperatures and urbanization could raise cloud base in the global set of TMCFs (Foster, 2001; Still et al., 1999; Williams et al., 2015), which could lead to some low-elevation TMCFs losing cloud immersion. In the Caribbean, the wet seasons are projected to be less wet by the end of the century (Karmalkar et al., 2013). Specifically, 1) shallow convection over Caribbean mountains in the early rain season may decrease if the trade winds continue to strengthen and shift direction, changing sea breeze dynamics and weakening orographic lift (Comarazamy and González, 2011); and 2) the regional late rain season may shorten due to strengthening of the Caribbean Low Level Jet (CLLJ) (Taylor et al., 2013).

The goal of this study was to determine cloud-base altitude and frequency of occurrence of low clouds during dry and wet seasons at a low-elevation tropical forest in the Luquillo Mountains, Puerto Rico, to establish a baseline against which to quantify future change. In addition, ceilometer, radiosonde, and satellite data for the northern Caribbean were analyzed to determine how the surrounding region influences the cloud patterns over ocean and land. Ceilometer data that are available to the research community are usually collected at airports, not in mountainous areas, and furthermore do not specifically quantify cloud frequency (Schulz et al., 2016). A dedicated ceilometer was installed to measure cloud base altitudes on the windward side of the mountains, at 100 m elevation and 7 km distance from the ridgeline at ~1000 m, and metrics were developed to quantify seasonal changes in base altitude and frequency of low, mountain-associated clouds. These metrics can also be used in the future to analyze temporal trends.

## 2 Study Area

The Luquillo Experimental Forest (LEF) is located in the northern tropics at 18.3° N, 65.7° W. The LEF area is in the Luquillo Mountains on the eastern end of the island of Puerto Rico; these mountains are no more than about 13 km from the coastline (the Atlantic Ocean to the north and east and the Caribbean Sea to the south). Maximum elevation is 1077 m. High rainfall causes steep, dissected slopes and the ridges and stream valleys are covered by undeveloped tropical forest. Weather and clouds in the Luquillo Mountains follow patterns typical of the Caribbean region, which is dominated by the easterly trade-winds (Malkus, 1955; Odum and Pigeon, 1970; Taylor and Alfaro, 2005). Annual trade-winds are driven by an interplay of the North Atlantic Subtropical High (NASH) sea level pressure system and the Inter-Tropical Convergence Zone (ITCZ) position (Giannini et al., 2000). In the northern hemisphere summer, the ITCZ moves to its northern position in the southwest Caribbean, weakening the trade-winds and giving way to the progression of tropical easterly low-pressure waves, which help to create a Caribbean wet season April/May through November (Gouirand et al., 2012). During winter the NASH extends westward, strengthening the trade-winds and suppressing convection to create a cooler dry season December through March (Gouirand et al., 2012). The CLLJ is a strengthening of the trade-winds in June and July, separating the rain season into early and late with a warm drier season called the mid-summer drought (Taylor et al., 2013). Trade-wind cumulus over the ocean are found within the trade-wind moist layer, limited above by the trade-wind inversion (TWI) and below by the lifting condensation level (LCL) (Stevens, 2005). Intra-annual variation in oceanic cloud cover is dampened as an estimated two-thirds of the cloud cover comes from annually-consistent clouds near the LCL; the other third is seasonally-changing clouds higher aloft (Nuijens et al., 2014). A pattern of lower clouds (bases and tops) over the ocean and higher clouds over the land, with a stronger diurnal effect of convection above land has been observed in the tropics with Cloud-Aerosol Lidar and Infrared Pathfinder Satellite Observation (CALIPSO) (Medeiros et al., 2010). However, with overpasses every 16 days, satellite data do not provide the level of temporal and spatial resolution required to effectively study the clouds at a TMCF (Costa-Surós et al., 2013). Satellite data in this study are used to generalize the ceilometer findings and quantify patterns of cloud top height and cloud thickness in the region.

Experiments and long-term monitoring in the Luquillo Mountains showed that orographic precipitation events occur consistently throughout the year (from the trade-winds and thermal lifting over the mountains), providing 25% of the rainfall by volume (Scholl and Murphy, 2014) as well as additional, unmeasured cloud water. Low clouds at the mountains have been hypothesized to originate with consistent winds carrying oceanic clouds to the mountains too rapidly for dissipation (Raga et al., 2016), and additional condensation may occur with topographic lifting of moist air up the slopes. The cloud forest in the Luquillo Mountains has been reported from near 600 m where epiphytes first occur in abundance and trees are noticeably smaller (Bruijnzeel et al., 2011; Weaver and Gould, 2013); however this elevation will vary with interactions between the topography and wind direction (e.g. the TMCF elevation range is almost certainly higher on the leeward slopes). It is possible the Luquillo Mountains may have already undergone cloud base lifting since the first field studies in 1963; when cloud immersion was observed at 500 m (Odum and Pigeon, 1970) and at 600 m as noted in Weaver (1995), but the evidence is anecdotal, and there were no cloud base measurements recorded over time. Seasonal patterns of clouds in the

Luquillo TMCF were studied with synoptic weather observations (Malkus, 1955; Odum and Pigeon, 1970) and more recently there have been summer cloud water deposition measurements (Eugster et al., 2006; Holwerda et al., 2011). Cloud base height data have been collected in the region by satellites and airport stations in addition to the newer data collected by this study nearer to the TMCF. These data types will be discussed extensively in the next section and a map of locations can 5 be seen in Fig. 4a.

**3 Methods**

3.1 Ceilometer and Weather Station Data in the Luquillo Experimental Forest

A Vaisala CL31 laser ceilometer installed at 100 m elevation 7 km northeast of the northwest-trending ridgeline that connects two of the higher peaks of the Luquillo Mountains. The ceilometer collected data at 30-second intervals from April 10 29, 2013 through August 1, 2016. Ceilometer data used in this study were the altitudes of the lowest cloud layer at a point above the instrument; the cloud layer base is the bottom of a vertically continuous layer at least 100 m thick with no vertical visibility (defined according to a 5% contrast threshold; http://www.vaisala.com). The mean and median cloud base values that are usually calculated from ceilometer data are not sufficient to answer our questions about the frequency of clouds immersing the relatively low-elevation forested mountains, because data from a point source include numerous clear-sky and 15 high cloud base observations. Therefore we developed metrics to summarize the cloud base altitude data in ways that could express differences and changes temporally (as well as spatially if the same metrics were calculated elsewhere) in cloud base for altitude range of interest in the TMCF; the forest is affected by clouds in the elevations from ~600-1077 m (Weaver, 1995). The metrics were derived from the frequency distributions of all cloud base measurements, where clear-sky was considered an infinite cloud altitude in the computations and clouds that may have been present below the elevation of the 20 ceilometer (100 m) were not recorded. The more traditional metric of 'average' would not be able to account for clear-sky observations. For example, if a day had one cloud in the 24-hour period, and it had a base of 200 m, the daily average would be 200 m. However, a day with half the cloud bases at 200 m and half at 2000 m would have a daily average cloud base of 1100 m, higher than the first day even though the second day had more clouds that would affect the TMCF. We did not trim the data to the altitude of interest; using all data gave a more complete picture of the cloud patterns of the troposphere above 25 the site.

The specific quantiles used for cloud-base metrics were chosen such that hourly metric values were between 600-1077 m a majority of the hours in each season and daily metric values were between 600-1077 m a majority of the days in each season. In this way, the metrics can be applied to help quantify the ecosystem characteristic low-elevation cloud amount needed to sustain the forest throughout the hour, day, and season. The metric 'hourly low cloud base' was defined as the altitude 30 marking the first quartile ($Q_1$) of the frequency distribution of the 120 measurements in each hour. The metric 'daily low cloud base' was defined at the daily first tertile ($T_1$) of the distribution of $Q_1$ altitudes from each hour, meaning the low cloud base metric was at or below that altitude for 8 (not necessarily consecutive) hours in the 24-hour day. The metric 'hourly minimum cloud base' was defined as the single-value altitude of the lowest cloud base detected during each hour. The metric

'daily minimum cloud base' was defined as the daily seventh octile ($O_7$) of the distribution of minimum values for each hour, meaning the minimum cloud base was at or below that altitude for 21 hours in the day. The daily low metric represents a cloud base altitude that is often observed in the portion of the day with the lowest clouds; whereas the daily minimum metric represents an altitude that lowest cloud base is found at throughout most of the day. For comparison with regional

Automated Surface Observation System (ASOS) data, the standard hourly average cloud bases and average cloud bases (omitting clear-sky data points) were also calculated at our Luquillo Experimental Forest site (LEF). Hourly cloud cover was computed as the fraction of positive cloud base detections, at any altitude, that occurred in the 120 instrument measurements per hour. An example of these calculations in practice and more information on the design of the metrics for this study and for other potential studies is given in the Supplement.

Hourly rainfall, relative humidity (RH), temperature, wind speed, and wind direction data were collected at several weather stations around the TMCF within an 8 km$^2$ area of the forest for differing periods of record (PORs) (Table 1). High hourly correlations were observed between these parameters across the forest stations for periods of overlap, giving confidence that patterns between the set of weather variables at each station in the immediate vicinity of the ceilometer measurements were homogeneous, although of differing magnitude at the different stations (Van Beusekom et al., 2015). Thus we used the

weather data from the one station with the most complete and longest POR (Bisley at 361 m; Table 1). Mean sea level pressure (MSLP) was only collected near the TMCF at 100 m. Its relatively short record was highly correlated with the MSLP at the ASOS station TJNR, Ceiba (correlation coefficient $\rho = 0.98$) , so the record was filled with data from Ceiba. The rainfall and RH data at 361 m and the MSLP at 100 m (missing values filled with data from Ceiba) were representative of seasonal patterns in the Luquillo Mountains and were used to interpret the ceilometer data by season locally and

regionally.

The 2000-2016 MSLP showed higher pressure in the dry seasons and lower pressure in the wet seasons (Fig. 1c), exemplifying the regional seasonal patterns that originate with the NASH and CLLJ (Brueck et al., 2014; Taylor et al., 2013). However, interannual variability in the timing of seasons from climate oscillations (North Atlantic, Pacific Decadal, and El Niño-Southern Oscillations) was observed in averages made over the shorter record (Gouirand et al., 2012). Rainfall,

RH, and MSLP months April 1-15 (M4.5), October (M10), and December (M12) in the LEF ceilometer period of record (LEF POR, May 2013 through August 2016) were not representative of the longer period 2000-2016 (Fig. 1). Because we have a short period of record for the ceilometer data (~3 years) and we focused on average seasonal behavior in this initial study, we chose to omit the unrepresentative time periods noted above and only used the time periods representative of the longer-term seasonal average (~17 years). As the ceilometer record gets longer, we will be better able to investigate effects

of the large-scale climate oscillations on clouds at the site.

All correlations between variables were computed as Pearson product-moment correlations with coefficient $\rho$ measuring linear correlation. Strengths of correlations were based on the relationships between weather parameters and clouds found in this study and another Caribbean cloud and weather study (Brueck et al., 2014); correlation was assumed fair if $0.3 < |\rho| < 0.4$, good if $0.4 < |\rho| < 0.6$, and very good if $|\rho| > 0.6$.

3.2 Ceilometer and Weather Station Data at Airports

The ASOS data for six airport stations in the Northern Caribbean from 2000-2016 were accessed from the Iowa Environmental Mesonet (Table 1), which include average hourly cloud base when clouds exist, cloud cover as percentage of the sky, and observed weather variables. No 30-second data were available and all data have been summarized to the mean-hour before reporting, thus not specifically quantifying cloud frequency. The ASOS network reports by a range of oktas (okta = 1/8 of the sky) covered by cloud (Nadolski, 1998): "clear" for 0, "few" for 1/8-2/8, "scattered" for 3/8-4/8, "broken" for 5/8-7/8, and "overcast" for 8/8. These numbers are automatically calculated by the instrument from 30-second detections similar to our calculations for the LEF site (Nadolski, 1998). For calculation of average POR cloud cover in this study, the cover categories were converted to sky cover fraction using the middle of the range or 0 for "clear", 0.125 for "few", 0.375 for "scattered", 0.75 for "broken", and 1 for "overcast".

To characterize variations in the cloud-system of the region, metrics were again developed to summarize the hourly average cloud base data at the ASOS and LEF stations in ways that expressed the changes in seasonal altitudes of clouds at altitudes meaningful for heights in the TMCF. The metric 'daily minimum mean-hour cloud base' was defined as the minimum value each day of the hourly averages and similarly the metric 'daily maximum mean-hour cloud base' was defined as the maximum of the hourly averages. The metric 'daily low mean-hour cloud base' was defined as the daily $Q_1$ of the hourly averages. To characterize the cloud-system at large, mean hourly cloud cover (in fraction of sky) was also calculated.

In order to represent usual atmospheric conditions, bearing in mind the data were mean-hours of existing cloud bases and not representing clear conditions that may have been present during the hour, hourly average cloud base was considered clear sky (in calculation of the daily metrics) if the cloud cover fraction was less than 50% of a station's POR mean cloud cover. POR mean cloud cover was 0.6 fraction of the sky at the LEF station (foothills) but 0.2-0.3 fraction of the sky at the ASOS stations (airports). Under this condition, ~2% of the data at each station were removed before calculation of 'usual' conditions.

3.3 Radiosonde Data at Airports

Two to three times daily, radiosonde profiles were available for 2000-2016 at Santo Domingo, San Juan, and Sint Maarten from the Integrated Global Radiosonde Archive (Durre et al., 2009). We expected to find cloud base altitudes above the LCL and cloud tops at or below the TWI for trade-wind conditions (Brueck et al., 2014; Malkus, 1955), thus we used the radiosonde profiles to compute values of minimum LCL and maximum TWI base over each day to quantify frequency of stable marine boundary layers and trade-wind cloud conditions at the site (Rastogi et al., 2016; Zhang et al., 2012). The LCL calculation we used involves an air parcel representing the mean potential energy in the lowest 50 hPa of the atmosphere for a mean-layer LCL; this accounts for the fact that the thermodynamic profile of the boundary layer may not be well-represented by that of the surface layer (Craven et al., 2002; for details see Appendix A). The TWI calculation used the

method of Cao et al. (2007) with the published pressure bounds for the Atlantic (Augstein et al., 1974; for details see Appendix A).

3.4 Satellite Data

CALIPSO satellite vertical feature mask data (Winker et al., 2009) 2006-2016 were analyzed for information on cloud tops and cloud thickness in the region. There are ten tracks in the region considered (Fig. 4a, Table 1); the northwest/southeast trending tracks are on the daytime orbit, and the other direction the nighttime orbit. Tracks are repeated at a 16 day interval and each satellite overpass records whether cloud was detected in 300 m horizontal by 30 m vertical cells up to 8.2 km altitude (data extend up to 30 km but we do not discuss those here). Data were summarized by fraction of measurements with cloud detections in larger cells of 0.05° latitude (approximately 6 km) with 30 m vertical resolution, as well as mean and $Q_1$ altitudes of the top and base of the lowest cloud layer over land and water, considering all regional tracks. None of the tracks went directly over the study site. As with the ceilometer metrics, the $Q_1$ altitudes summarize the low cloud frequency behavior in a way that can express differences and changes temporally as well as spatially. Aproximately 10% of the vertical profiles had uncertainty in the cloud base as the lidar was completely attenuated below a cell of cloud (because the cloud was too thick optically; Winker et al., 2009). For these profiles, low cloud base altitude was set as minimally the altitude of land/water surface, and maximally as the lowest cell vertically that cloud is observed, before complete lidar attenuation; this results in a range reported for the low cloud base altitude and layer thickness.

**4 Results**

4.1 Ceilometer and Weather Station Results in the Luquillo Experimental Forest

The data suggest that the TMCF has a greater probability of cloud interaction during the mid-summer dry season (MSD) throughout the elevation range of the forest between 600-1000 m, and during the winter dry season at the upper elevational reaches (<1000 m), than it does in the wet seasons (Figs 2, 3a). Of all the seasons, the mid-summer dry season (Fig. 2c) had low and minimum cloud bases at 600-750 m most consistently. The cloud base levels in the winter dry season (Fig. 2a) were not below 750 m as often as in the wet seasons, but were also absent above 3000 m, so on an hourly and daily basis, the low and minimum cloud bases were below 1000 m more consistently in the dry than in the wet seasons (summing the bars in Fig. 2). During the daytime hours (09:00-18:00) the median hourly low and minimum cloud bases were as low or lower in altitude in the dry seasons as the wet seasons (Fig. 3a); this was also the time of day when rainfall was the least frequent in the dry seasons (Fig. 3b). The means of median-hourly low and minimum cloud bases over the entire POR (the data shown in Fig. 3a) were 915 m and 702 m, respectively.

Of all available weather variables, the daily low and minimum cloud bases correlated best with RH and MSLP (negative correlations, Table 2, Figs 1a, c). Interestingly, RH did not correlate well with rainfall (Table 2) or with pressure indicators of wet and dry seasons (visually confirmed in Fig. 1b).

4.2 Ceilometer and Weather Station Data at Airports

In the northern Caribbean region, the mean low cloud base was lower during the mid-summer dry season than during the other seasons at all of the stations, indicating seasonal variation (Fig. 4c). However, the longer winter dry season had a higher low cloud base than the wet seasons (Fig. 4b) except at the LEF station (the station in the Luquillo foothills; note at this station in Fig. 4b, less frequent winter dry season high clouds (see Fig. 1) lowers the mean-hour cloud base). Cloud cover percentage was highest in the mid-summer dry season relative to the other seasons at all the stations to the east (windward) and including the LEF station, and winter dry season cloud cover was similar to or higher than the wet seasons (Fig. 4d). The low and minimum cloud bases for stations near the Luquillo Mountains correlated much better with cloud cover than at the stations as a whole, with very good negative correlation at the LEF station itself (Table 2, S3).

In contrast to the LEF area (Fig. 1b) the regional pattern of RH for 2000-2016 was as expected, with lower RH during the dry seasons than the wet seasons (Fig. 5) and the low and minimum cloud bases all had fair correlation with the RH measurements (Table 3). Regionally, no metrics of daily clouds correlated with MSLP except at the LEF site, where the low and maximum cloud bases had good and very good correlation with MSLP, respectively (Table 3). The regional means and correlations were also computed over the shorter LEF POR with exclusion of the non-representative months. These were not substantially different than the 2000-2016 results presented here (in Fig. 4, and Fig. 5, Table 3), but the mathematical strength of the resulting means and significance of the correlations necessarily decreased with less data.

4.3 Radiosonde Data at Airports

Over the North Caribbean regional ceilometers excluding the LEF site, the maximum cloud base altitudes were below the TWI and the lowest near the mean-layer lifting condensation level (MLLCL) (both calculated from 2000-2016 radiosonde profiles, Fig. 4b). The TWI was distinguishable 86% of the time in the winter dry season, 74% in the mid-summer dry season, 71% in the early rain season, and 65% in the late rain season. The averages of maximum daily TWI values were 1921 m for Santo Domingo, 2141 m for San Juan, and 1851 m for Sint Maarten, with average seasonal standard deviation of ±48 m and the lowest TWI altitudes in the mid-summer dry season. This agrees with other research on global distribution and seasonal variance (Guo et al., 2011), and the influence of local topography (Carrillo et al., 2015), on the TWI. The averages of minimum daily MLLCL values were 538 m for Santo Domingo and 557 m for San Juan, with average seasonal standard deviation of ±27 m and the lowest values in the mid-summer dry season.. At Sint Maarten, the POR-ceilometer-observed mean daily minimum cloud base was 202 m lower than the calculated minimum daily average MLLCL of 705 m, whereas at Santo Domingo and San Juan the difference was 37 m lower and 34 m higher, respectively. Consequently, we assumed the radiosonde measurements were not representative of conditions above the ceilometer location at Sint Maarten, and these data were not used in the regional comparison.

4.4 Satellite Data

CALIPSO data from ten tracks over the northern Caribbean (Fig. 4a) showed that land with high topography increased the likelihood of clouds in a layer from land surface elevation up to 2 km. The 2 km altitude is approximately the level of the

TWI, which under most conditions caps the trade-wind cumuli (Fig. 6). Oceanic clouds were most likely to be found in a layer from 400 m to 1 km. Lidar attenuation causes uncertainty in the cloud base identification for the thickest 10% of clouds, as discussed earlier. For the lowest cloud layer below 8.2 km on all ten tracks, only considering when clouds are present, the mean altitude of the base of the layer ranged from 659-1385 m over ocean and 796-1155 m over land, and the

mean altitude of the top of the layer was 2167 m over ocean and 2034 m over land. This indicated a similar cloud layer altitude over ocean and land; most land in the tracks was low topography so this is not unexpected. However, considering all data (i.e. clear-sky has infinite cloud base and top altitude), the $Q_1$ altitude range (from uncertainty due to lidar attenuation) of the base of the layer was 235-475 m over ocean and 25-355 m over land, and the $Q_1$ altitude of the top was 1165 m over ocean and 1315 m over land. Thus, there were possibly more low-altitude cloud base observations over land, depending on

where in the range the cloud base actually existed. Highest topography land (Fig. 6c) exhibited an upper limit of the cloud layer around 2 km, however lower topography land had thicker clouds than over ocean. The mean layer thickness was 442-853 m over ocean and 582-818 m over land. Thicker clouds with more low bases occurring over land has also been found in previous studies looking at ocean and low topography land at different locations and in larger regions (Medeiros et al., 2010; Rauber et al., 2007).

**5 Discussion and Conclusions**

Caribbean trade-wind cumulus clouds have been observed in other studies to be relatively shallow and originating with bases near the LCL (Malkus, 1955; Nuijens et al., 2014; Rauber et al., 2007; Zhang et al., 2012). The regional ASOS and CALIPSO data in this study support the previous findings, with most oceanic cloud bases at or within 100 m of the LCL (Figs 4 and 6). Both 5-day and monthly correlations between low clouds and RH were observed region-wide (Table 3). It has

been theorized that a stronger TWI increases oceanic cloud cover and lowers the cloud base, and cloud amounts reflect the upstream boundary-layer atmospheric conditions (Myers and Norris, 2013; Stevens, 2005). The Luquillo Mountains are ~10 km from the coast, and trade-wind velocity in the region is typically 3-5 m/s (Lawton et al., 2001). Existing clouds are over land for about 30 minutes before reaching the TMCF, and orographic lifting may generate additional condensation during parts of the daily cycle. We postulate that proximity of the TMCF to the coast is an ecological advantage in that the forest

has adapted to the oceanic atmosphere (relatively invariant humidity, temperature profile, and cloud cover). The Caribbean ASOS stations east (windward) of Puerto Rico on small, low-relief islands may be influenced by the thermodynamically stable marine boundary layer TWI and reflect conditions closer to those over the ocean surface than to those on land. These stations have higher cloud cover in the dry seasons (Fig 3d) and lower clouds correlating with increasing cloud cover (Table 3); remnants of this weather pattern may be persisting on to the land in eastern Puerto Rico (Raga et al., 2016).

However, our results clearly show the changes that occur when the clouds interact with land that extends above the LCL in a trade-wind regime: cloud base altitude and cloud cover over the mountains was increased compared to that over small low-relief islands or open ocean (Figs 4 and 6). In addition, the cloud base over land rises high enough that a TWI limit to vertical extent of the clouds may result in shallower (thinner) cloud layers at the highest land elevations (Fig. 6) (Rastogi et

al., 2016). When topography is close to the ocean this effect is especially pronounced, as seen in the CALIPSO data with the disproportionate effect on the clouds by the lower peak elevation but closer oceanic proximity of the northernmost peninsula of the Dominican Republic Fig 6a, b). The local ceilometer data showed that the correlation of daily low and minimum cloud base metrics with RH was primarily driven by cloud bases lower than 1000 m in the calculations, and correlation of these

metrics with MSLP was primarily driven by cloud bases higher than 1500 m in the calculations (Table 2). In addition, the relatively high RH during the dry seasons in the mountains is a pattern that is not seen over the low-relief sites (Figs 1b, 5) highlighting the importance of site-specific measurements of parameters that influence cloud base for TMCFs, rather than relying on data from regional stations at airports.

The Caribbean region is projected to undergo drying in the future, with various mechanisms that will suppress deep

convection and affect the frequency of high rainfall (Karmalkar et al., 2013). Low peak-elevation TMCFs are especially vulnerable in a changing climate, because slight increases in cloud base altitude could end cloud immersion for the entire forest (Foster, 2001; Lawton et al., 2001; Ray et al., 2006). With our results, we question the likelihood of frequent cloud base at 600 m at the Luquillo Mountains (Figs 2, 3), the published TMCF lowest elevation based on older observations of cloud immersion (Odum and Pigeon, 1970; Weaver, 1995) and observed changes to smaller trees which dominate cloud

forests (Weaver and Gould, 2013). It is possible that such characteristic vegetation may persist for some time after changing climatic conditions have lifted the cloud base (Oliveira et al., 2014; Richardson et al., 2003).  Rainfall and relative humidity in the forest are currently quite high so that some species, once established, may be able to survive if rainfall remains high (Martin et al., 2011), but the diversity and numbers of epiphytes and other species that depend on cloud immersion might decline over time. During a past wet-season drought at Luquilo, trade-wind precipitation became very important in the

absence of deep convection (Clark et al., 2017). If trade-wind cloud layers become thinner and shallow convection weakens, drought effects on the TMCF could be even more significant.

This study presents evidence that cloud levels in the dry season are consistently as low, or lower, than in the wet seasons at a low-elevation TMCF under the current climate regime, indicating that the TMCF ecosystem may be more vulnerable to wet-season drought periods than was previously assumed. We calculated the mean MLLCL at 557 m and mean TWI of 2141 m

from San Juan airport radiosonde data, identifying the local trade-wind cloud layer near Puerto Rico. While previous studies report the forest clouds to start at 600 m, our ceilometer observations over the forest about 7 km windward of the TMCF showed the lowest cloud bases most frequently occurred at higher elevation, from 702 m to 915 m. Continued long-term measurements at the Luquillo Mountains, and comparisons to other low-elevation TMCFs, are needed to expand the low cloud pattern temporally and geographically with full confidence, and to determine how projected changes in regional

temperature and atmospheric circulation patterns will affect these cloud-water-dependent ecosystems.

## 6 Data Availability

The Luquillo Experimental Forest ceilometer and weather station data is hosted on the Critical Zone Observatory Data Portal http://criticalzone.org/luquillo/data.

## Appendix A

This Appendix includes further detail on the calculation of the lifting condensation level (LCL) and the trade-wind inversion (TWI).

The LCL is the altitude at which a parcel of air lifts adiabatically and cools to the temperature that its relative humidity is 100%. Assuming the presence of cloud condensation nuclei, condensation of water vapor into liquid water cloud droplets is possible at and above this altitude. If the boundary layer is well-mixed, the layer will have a constant potential temperature

and a constant mixing ratio and air parcel parameters at any altitude can be used to calculate the LCL; these weather parameters are most often measured at the land surface. However, the mean-layer (ML)LCL has been shown to be a more accurate descriptor of the lower boundary of cloud development than the surface-parcel based LCL (Craven et al., 2002). An MLLCL was determined from each radiosonde profile by first splining the temperature, dewpoint temperature, and pressure measurements into 5 hPa levels and calculating potential temperature and water vapour mixing ratio at each level (following

Bolton, 1980). Then the mean potential temperature and water vapour mixing ratio in the lowest 50 hPa of the atmosphere was computed to define the mean-layer parcel. The pressure-temperature intersection of the mean-layer parcel potential temperature isopleth with the mixing ratio isopleth defined the MLLCL. The difference between the mean-layer parcel surface temperature and MLLCL temperature was used with the dry adiabatic lapse rate to calculate the LCL altitude.

The TWI is a layer often present in the atmosphere in the trade-wind latitudes characterized by a reversal of the usual

negative temperature gradient to a positive gradient of warming with increasing altitude. A TWI base was determined from each radiosonde profile following published methods (Cao et al., 2007). The ascending radiosonde instruments recorded measurements at uneven intervals of roughly 50-300 m at the altitudes and time periods of interest to this study. The bounding altitudes containing the TWI base were determined by setting the upper bound as the lowest altitude observation with positive temperature gradient $dT_p/dz$ between itself and the next highest observation, greater than 0 °C temperature and

0% RH, with pressure between 70 and 95 kPa. The pressure bounds are for the Atlantic Ocean TWI (Augstein et al., 1974). The lower bounding altitude was set as the observation one below the upper bounding altitude, which necessarily had a negative temperature gradient $dT_n/dz$ below it. Inside the bounding altitude layer, the temperature gradients were then assumed to equal the positive gradient $dT_p/dz$ above the TWI base and the negative gradient $dT_n/dz$ below the TWI base. The intersection of these two lines of constant positive and negative gradient inside the bounding layer was set as the TWI base

altitude.

*Acknowledgements.* This research was supported by the Luquillo Critical Zone Observatory (EAR-1331841) and Grant DEB 1239764 from the U.S. National Science Foundation to the Institute for Tropical Ecosystem Studies, University of Puerto Rico, and to the International Institute of Tropical Forestry (IITF) USDA Forest Service, as part of the Luquillo Long-Term Ecological Research Program. The U.S. Forest Service (Department of Agriculture) Research Unit, the U.S. Geological Survey Climate and Land Use Change WEBB

Program, and the University of Puerto Rico gave additional support. Samuel Moya, Carlos Estrada, and Carlos Torrens provided field assistance. William A. Gould, Ariel E. Lugo, Paul W. Miller, and Sheila F. Murphy provided comments on the manuscript, and we thank the anonymous reviewers whose comments improved the paper. Any use of trade, product, or firms names is for descriptive purposes only and does not imply endorsement by the U.S. Government.

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

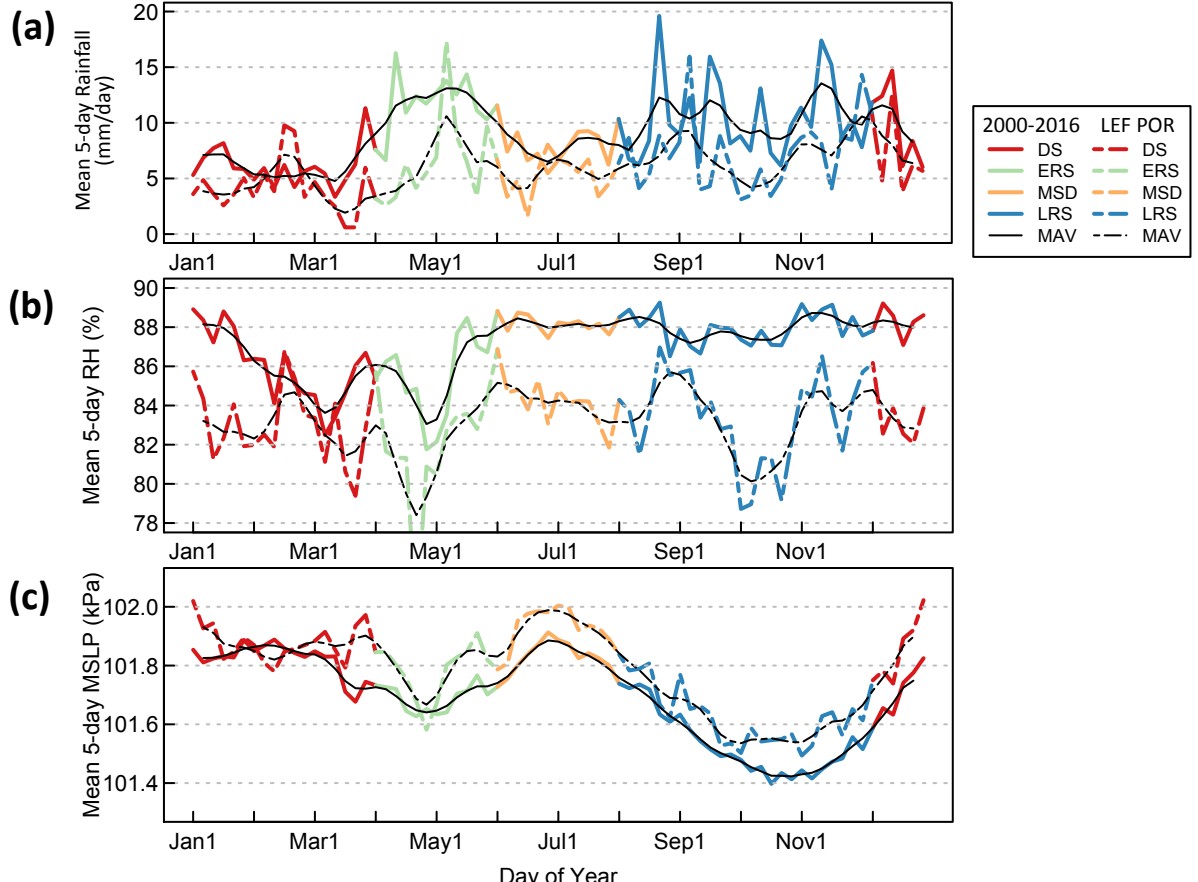

**Figure 1: Day of year mean values for: a) precipitation at 361 m; b) relative humidity (RH) at 361 m; and c) mean sea level pressure (MSLP) at 100 m in the Luquillo Experimental Forest (LEF) sites . The colors show the 5-day means during the winter dry season (DS), early rain season (ERS), mid-summer dry season (MSD), and late rain season (LRS). The black line is the 15-day moving average (MAV), with the 2000-2016 period as solid lines and the LEF ceilometer period of record (LEF POR) of May 2013 through August 2016 as dashed lines.**

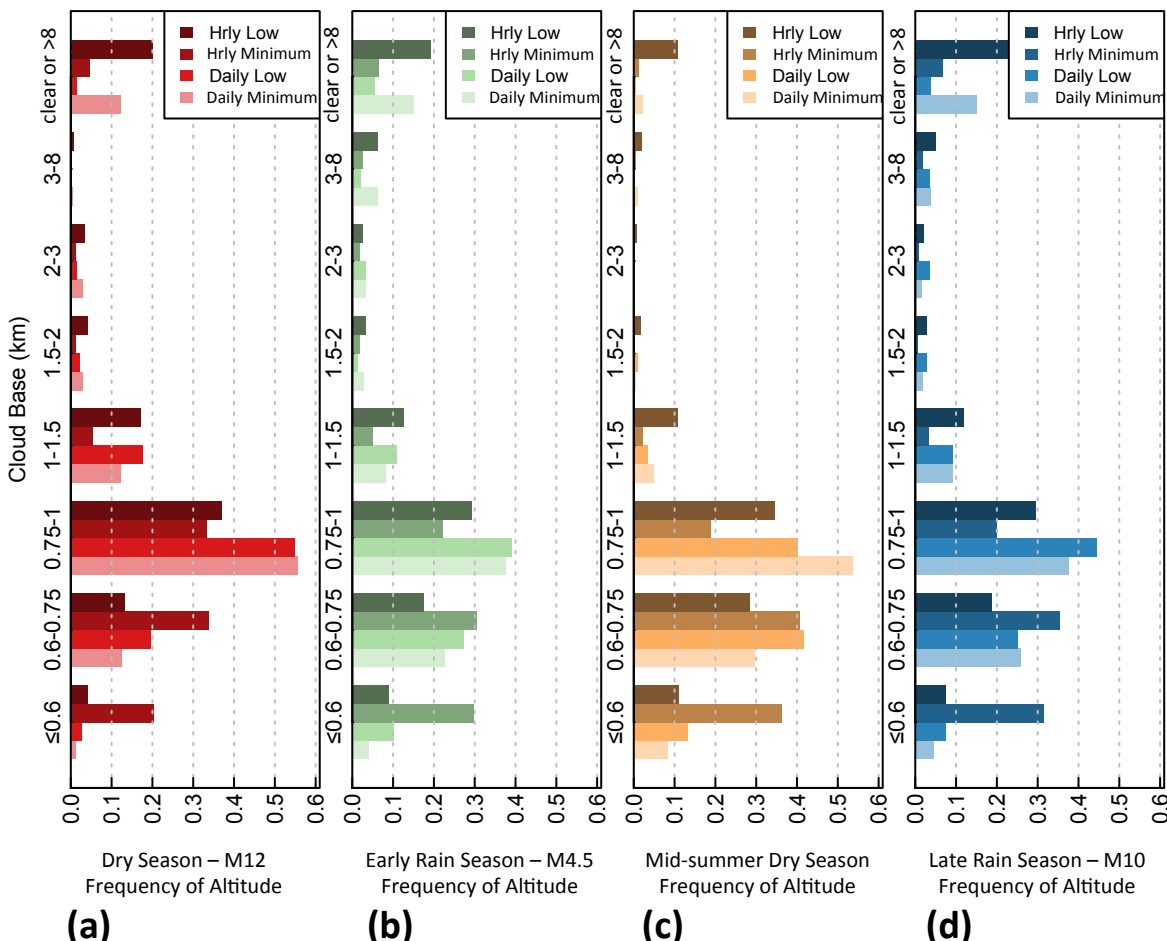

**Figure 2: Frequency of low cloud base by season excluding non-representative months in each year from the Luquillo Experimental Forest ceilometer, May 2013 through August 2016: a) dry season without month December (M12); b) early rain season without April 1-15 (M4.5); c) mid-summer dry season; d) late rain season without October (M10). The metrics are hourly low cloud base (first quartile ($Q_1$) of each hour of 30-second cloud base altitudes), hourly minimum value, daily low value (first tertile ($T_1$) of the hourly low set), and daily minimum value (seventh octile ($O_7$) of the hourly minimum set).**

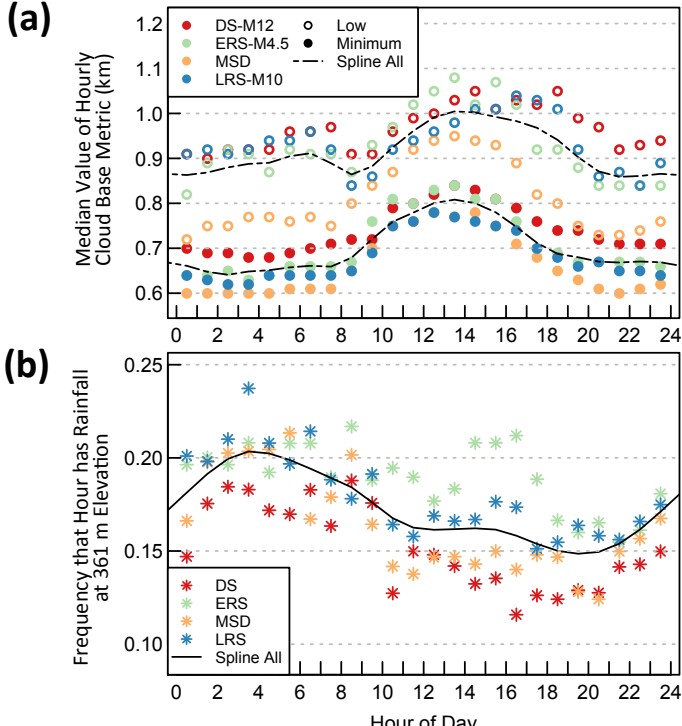

**Figure 3: Summaries by hour of day for each season: a) median altitude of hourly low cloud bases and median altitude of hourly minimum cloud bases from the Luquillo Experimental Forest ceilometer, May 2013 through August 2016, for the winter dry season without month December (DS – M12), the early rain season without April 1-15 (ERS – M4.5), the mid-summer dry season (MSD), and the late rain season without October (LRS – M10); b) frequency measureable rainfall occurred in each hour of the day for 2000-2016 for the Bisley station at 361 m elevation during the seasons without excluded months. Colored symbols are seasonal values and black lines are the cubic smoothing splines of all data in the ceilometer period of record.**

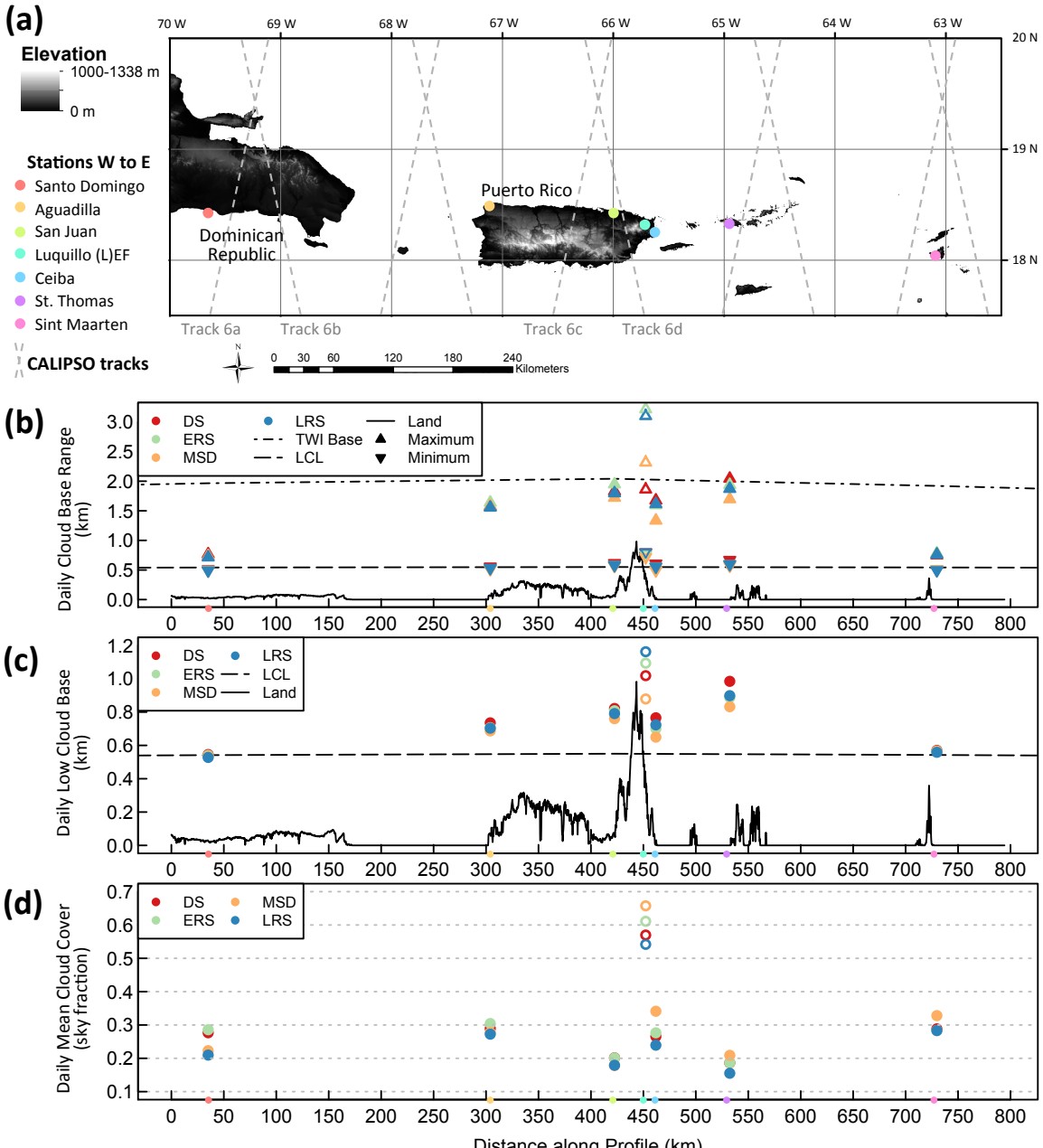

**Figure 4. Daily 'usual' mean clouds averaged over the winter dry season (DS), early rain season (ERS), mid-summer dry season (MSD), and late rain season (LRS) for the period of record (POR) across the region: a) locations of airport (ASOS) stations, Luquillo Experimental Forest (LEF) station, and CALIPSO satellite tracks, along with topography; b) daily minimum and maximum cloud bases (of hourly means), splined lines of trade wind inversion (TWI) base and lifting condensation level (LCL) based on three geographical calculations, and land surface elevation along profile; c) daily low cloud base (first quartile (Q₁) of hourly means), estimated LCL, and land surface elevation along profile; d) daily mean cloud cover as fraction of sky. The land surface profile trends due east and west (shifting north and south to be near the stations). Station locations are marked with circles on the x-axis in the same colors as plot a. Dashed colored lines between San Juan and Ceiba are used to indicate that the LEF station measurements are only based on the available LEF POR, May 2013-August 2016 excluding non-representative months the entire 2000-2016. 'Usual' defined as hours with cloud cover at least 50% of mean-POR cloud cover for station.**

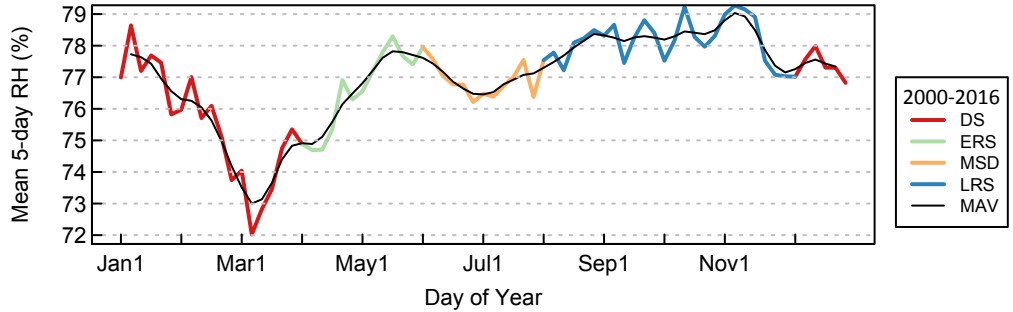

**Figure 5. Day of year mean values for relative humidity (RH) averaged over all 6 regional airport stations (ASOS, does not include Luquillo Experimental Forest data). The colors show the 5-day means during the winter dry season (DS), early rain season (ERS), mid-summer dry season (MSD), and late rain season (LRS). The black line is the 15-day moving average (MAV) for the 2000-2016.**

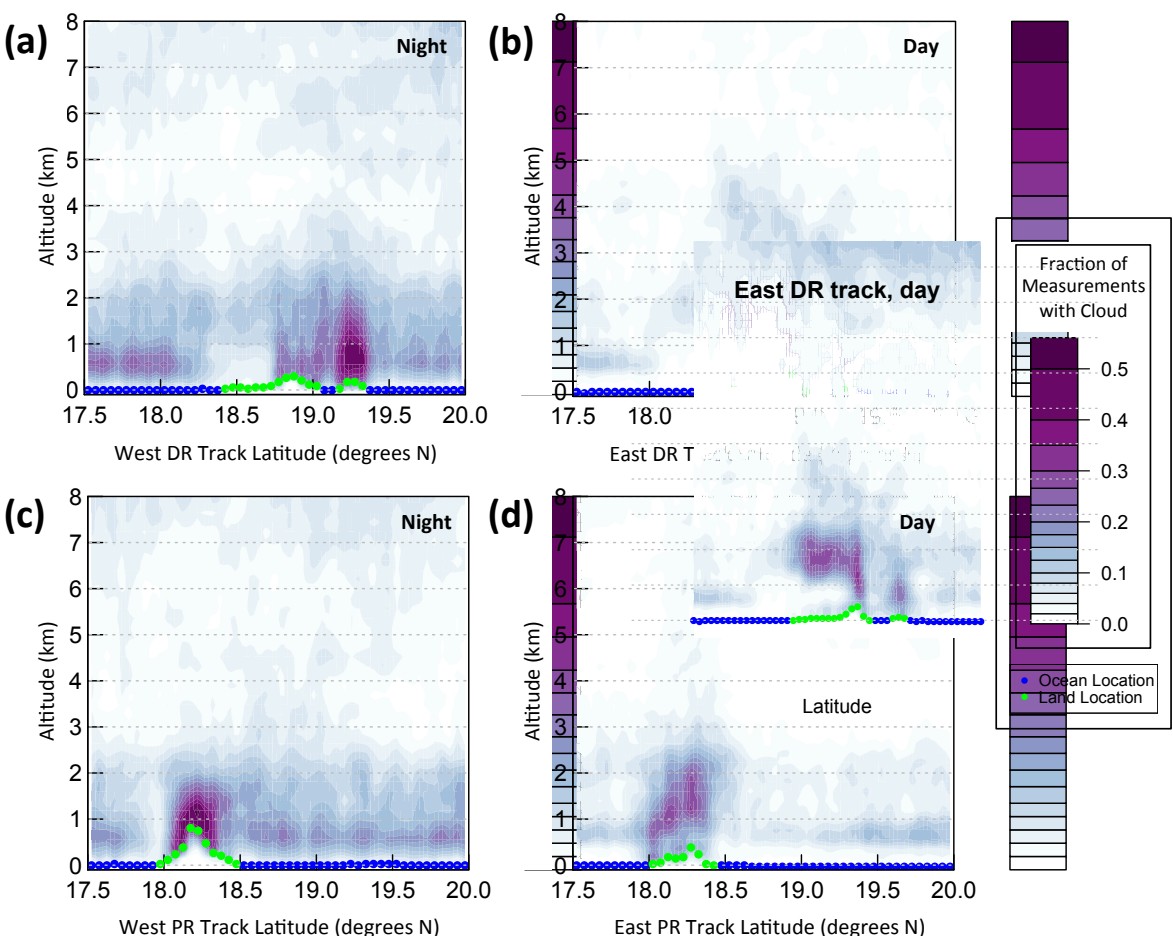

**Figure 6.** CALIPSO satellite data fraction of measurements with cloud recorded in cell, for years 2006-2016 with cell as 0.05° latitude along track and 30 m altitude for: a) West Dominican Republic (DR) track (night orbit); b) East DR track (day orbit); c) West Puerto Rico (PR) track (night orbit); and d) East PR track (day orbit). Since track measurement location varies slightly, surface point (land or ocean) is plotted as the mean location. Cell fractions are spline-smoothed into a surface. Tracks are marked on Figure 4a.

| Station | Lat (° N) | Long (° W) | Elev (m) | Land Cover | Variables[‡] | Period Used | Data Missing | Data Reference |
|---|---|---|---|---|---|---|---|---|
| Sabana LEF[*†] | 18.32 | 65.73 | 100 | Forest | RF, RH, MSLP, T, WD, WS | 5/2013 – 8/2016 | 5% | http://criticalzone.org/luquillo/data/ |
| Bisley LEF[*] | 18.30 | 65.75 | 361 | Forest | RF, RH, T, WD, WS | 1/2000 – 8/2016 | 3% | http://criticalzone.org/luquillo/data/ |
| Icacos LEF[*] | 18.28 | 65.79 | 645 | TMCF | RF | 1/2000 – 1/2016 | 7% | 50075000 http://waterdata.usgs.gov/nwis |
| Santo Domingo[†] | 18.43 | 69.67 | 18 | Airport | RH, T, WD, WS, S | 1/2000 – 8/2016 | 28% | MDSD http://mesonet.agron.iastate.edu |
| Aquadilla[†] | 18.49 | 67.13 | 72 | Airport | RH, T, WD, WS | 1/2000 – 8/2016 | 35% | TJBQ  http://mesonet.agron.iastate.edu |
| San Juan[†] | 18.43 | 66.01 | 3 | Airport | RF, RH, MSLP, T, WD, WS, S | 1/2000 – 8/2016 | 1% | TJSJ   http://mesonet.agron.iastate.edu |
| Ceiba[†] | 18.26 | 65.64 | 12 | Airport | RF, RH, MSLP, T, WD, WS | 1/2000 – 8/2016 | 37% | TJNR http://mesonet.agron.iastate.edu |
| St. Thomas[†] | 18.34 | 64.97 | 7 | Airport | RF, RH, MSLP, T, WD, WS | 1/2000 – 8/2016 | 6% | TIST    http://mesonet.agron.iastate.edu |
| Sint Maarten[†] | 18.04 | 63.11 | 4 | Airport | RH, T, WD, WS, S | 1/2000 – 8/2016 | 7% | TISM   http://mesonet.agron.iastate.edu |
| CALIPSO tracks[†] | 17.5-20 | 62.5-70 | vary | Various | Cloud vertical profiles | 6/2006 – 8/2016 | 0.05% | http://www-calipso.larc.nasa.gov |

[*] In Luquillo Experimental Forest (LEF)

[†] Cloud data collected at these stations

[‡] Key to abbreviations: RF = rainfall, RH = relative humidity, MSLP = mean sea level pressure, T = temperature, WD = wind direction, WS = wind speed, S = radiosonde

**Table 1. Data Record Information.**

| Variable | Cloud Base Altitude | | | | Rainfall | |
|---|---|---|---|---|---|---|
| | Daily Low (Daily $T_1$ of $Q_1$ of each hour) | | Daily Minimum (Daily $O_7$ of min of each hour) | | | |
| | 5-day $\rho$ | Month $\rho$ | 5-day $\rho$ | Month $\rho$ | 5-day $\rho$ | Month $\rho$ |
| RH | **-0.34** | -0.31 | **-0.36** | -0.28 | **0.13** | *0.05* |
| MSLP | **-0.45** | **-0.58** | **-0.52** | **-0.54** | **-0.30** | **-0.46** |
| Cloud Cover | **-0.69** | **-0.76** | **-0.72** | **-0.75** | **0.26** | *0.21* |

Significant relationships indicated by typeface: p-value < 0.1, p-value < 0.05, **p-value < 0.01**

Insignificant relationships *italicized*.

**Table 2. Correlations Coefficients $\rho$ of Relative Humidity (RH), Mean Sea Level Pressure (MSLP), and Cloud Cover with Cloud Base Altitude Metrics and Rainfall in the Luquillo Experimental Forest.**

| Variable | Cloud Base Altitude | | | | Rainfall | |
|---|---|---|---|---|---|---|
| | Daily Low (Daily $T_1$ of $Q_1$ of each hour) | | Daily Minimum (Daily $O_7$ of min of each hour) | | | |
| | 5-day $\rho$ | Month $\rho$ | 5-day $\rho$ | Month $\rho$ | 5-day $\rho$ | Month $\rho$ |

| Variable | Station | Cloud Base Altitude | | | | | | Rainfall | |
|---|---|---|---|---|---|---|---|---|---|
| | | Daily Low Mean-Hour (Q$_1$ of usual hrly means) | | Daily Minimum Mean-Hour (min of usual hrly means) | | Daily Maximum Mean-Hour (max of usual hrly means) | | | |
| | | 5-day ρ | Month ρ | 5-day ρ | Month ρ | 5-day ρ | Month ρ | 5-day ρ | Month ρ |
| RH | ASOS all[*] | **-0.35** | **-0.30** | -0.36 | *-0.26* | *-0.06* | *-0.17* | NA | NA |
| | ASOS near[†] | **-0.36** | **-0.29** | **-0.47** | *-0.33* | *-0.01* | *-0.08* | **0.22** | 0.17 |
| | LEF[‡] | **-0.22** | *-0.19* | **-0.37** | -0.36 | *0.04* | *-0.01* | **0.13** | *0.05* |
| MSLP | ASOS all[*] | NA | NA | NA | NA | NA | NA | NA | NA |
| | ASOS near[†] | *0.04* | *0.01* | **0.17** | 0.14 | -0.10 | *-0.08* | **-0.27** | **-0.42** |
| | LEF[‡] | **-0.40** | **-0.64** | -0.16 | -0.30 | **-0.46** | **-0.57** | **-0.30** | **-0.46** |
| Cloud Cover | ASOS all[*] | -0.21 | *-0.18* | **-0.18** | *-0.08* | *0.02* | *0.01* | NA | NA |
| | ASOS near[†] | **-0.30** | **-0.42** | **-0.27** | **-0.28** | *-0.03* | *-0.14* | *-0.01* | *-0.10* |
| | LEF[‡] | **-0.46** | **-0.76** | **-0.47** | **-0.82** | *-0.02* | -0.37 | **0.26** | *0.21* |

Significant relationships indicated by typeface: p-value < 0.1, p-value < 0.05, **p-value <0.01**

Insignificant relationships *italicized*.

[*] Mean of ρ from all six airport (ASOS) stations 2000-2015

[†] Mean of ρ from three airport (ASOS) stations nearest to Luquillo 2000-2015: San Juan, Ceiba, St. Thomas

[‡] ρ from Luquillo Experimental Forest (LEF) stations 5/2013 – 12/2015

'Usual' defined as hours with cloud cover at least 50% of mean-POR cloud cover for station.

**Table 3. Correlation Coefficients ρ of Relative Humidity (RH), Mean Sea Level Pressure (MSLP), and Cloud Cover with Cloud Base Altitude Metrics and Rainfall Region-Wide.**