# Peer review of "Analyzing cloud base at local and regional scales to understand tropical montane cloud forest vulnerability to climate change"

_Atmospheric Chemistry and Physics, 2016_

## Referee Comment (RC1) · Anonymous Referee #2 · 7 Feb 2017

Review of acp-2016-1166: "Analyzing cloud base at local and regional scales to understand tropical montane cloud forest vulnerability to climate change"

This paper describes results of an extensive cloud and meteorological monitoring campaign in the Caribbean in order to better understand the distribution of low boundary-layer marine clouds in time and space. Monitoring was uniquely carried out in mountain regions to gain insights about cloud immersion of tropical montane cloud forests and establish baseline conditions against which future changes may be measured. There are many results that I believe most readers (including me) will find cumbersome to wade through, but that is the nature of such a paper that reports on an extensive field campaign and important general conclusions were still clear. Low clouds are frequent

in the year round, with the highest frequency occurring in the dry seasons. Spatial distributions of cloud frequency and base height were quite spatially heterogeneous, however, likely due to topographic effects. Although it is commonly believed that tropical montane cloud forests would be most sensitive to changes in the frequency and elevation of dry-season clouds, authors interpret their detection of a common presence of wet season clouds to indicate that changes to these clouds could also be ecologically impactful. This paper describes critical ground-work and initial results necessary for long-term monitoring of cloud variations and changes in a unique and ecologically important zone. The complicated nature of the results is just the nature of the beast for exploratory studies such as this one that include multiple sites, multiple metrics for measuring mean cloud height, and multiple monitoring tools test for corroboration. I therefore believe this paper is appropriate for publication in ACP.

Specific comments:

P1 L29: It is not explicitly clear what "higher" is relative to.

P2 L2: I would also imagine fog water input varies greatly throughout a forest depending on they types of surfaces available for collection and micro-climatic wind patterns.

P2 L20-22: It is worth mentioning a couple other relevant studies that have used long-term cloud-base observations to investigate spatiotemporal variations and trends in cloud frequency and cloud-base height. Williams et al. (2015, GRL) used records from 24 airfields in southern California to show that substantial warming-induced increases in cloud base have corresponded to local land-cover change via nighttime urban warming. Richardson et al. (2003, J Climate) analyzed cloud heights at 24 airfields in the Appalachian Mountain region of the eastern US and documented multi-decade (1973-1999) increases (decreases) in cloud-base height north (south) of 37.5°N. Causes of the changes are not rigorously investigated in that study but the ecological implications are stressed as important. Further, Rastogi et al. (2015, Earth Interactions) presented a relevant study that combined cloud-base observations with satellite and radiosonde

observations to estimate how marine-layer stratus clouds intersect with topography.

Rastogi, B., A. P. Williams, D. T. Fischer, S. Iacobellis, K. McEachern, L. M. V. Carvalho, C. Jones, S. A. Baguskas, C. J. Still (2015), Spatial and temporal patterns of cloud cover and fog inundation in coastal California: ecological implications, Earth Interactions, In Review.

Richardson, A. D., E. G. Denny, T. G. Siccama, X. Lee (2003), Evidence for a rising cloud ceiling in eastern North America, Journal of Climate, 16(12), 2093-2098, doi:10.1175/1520-0442(2003)016<2093:EFARCC>2.0.CO;2.

Williams, A. P., R. E. Schwartz, S. Iacobellis, R. Seager, B. I. Cook, C. J. Still, G. Husak, J. Michaelsen (2015), Urbanization causes increased cloud-base height and decreased fog in coastal southern California, Geophysical Research Letters, 42(5), 1527-1536, doi:10.1002/2015GL063266.

There are many non-traditional abbreviations that I find distracting. For example: ERS, LRS, DS, MSD, TWI. By the half-way point of page 3 I'm fearing that at some point the entire paper may be abbreviated. I recommend unabbreviating some or all of the abbreviations that I list above and I believe doing so will enhance the interpretability and impact of the paper.

P4 L2-3: I think there is a word missing from this sentence.

P4 L28-30: It is not indicated what the time-step is for the values considered in this correlation analysis. Daily or annual cycles would cause correlation even if variability that is independent of the cycles is not correlated.

---

## Referee Comment (RC2) · D. Baumgardner (Referee) · 24 Feb 2017

This manuscript is a revision of a submission that I reviewed prior to its publication as a discussion. I am pleased that the authors acted on some of my recommendations with respect to expanding its scope and supplementing the ceilometer measurements with satellite and radiosonde data. In my opinion the current submission is much improved and I recommend publication. The results are compelling and provide useful information about clouds that impact Tropical Montane Forests.

The only recommendation that I have is to add more explanation about the metrics that are used in defining the cloud bases. I would suggest making this part of the Appendix. A representative frequency histogram of cloud bases, such referred in the text, could

be used to mark the various quartiles and tertiles that are being used to define cloud base minimums. In this same appendix it would be useful to further explain to the reader why these particular metrics are optimum, as there are no references to other studies that might have used them. If this is a first time such metrics have been used, then future studies by other investigators would have a cite-able reference if they use the same metrics.

---

## Referee Comment (RC3) · Anonymous Referee #3 · 1 Mar 2017

This paper describes measurements of cloud base altitude using ground-based remote sensing at a site in close proximity to the Luquillo Mountains, Puerto Rico. The authors stress the importance of cloud immersion to the current ecosystem of the tropical montane cloud forest, while also advising the reader that the various mechanisms by which these ecosystems exchange water with the atmosphere are still not fully understood. The authors suggest that rising cloud base altitude as a result of climate change would stress vulnerable species even if rainfall rates in these regions were to remain high. In addition, they hypothesize that the lack of cloud water deposition and increased evapotranspiration, resulting from elevating cloud base, could affect the watershed dynamics in the mountains. The authors also project that their study site may also be vulnerable

to changes in the wet season, especially during wet season drought periods.

While the manuscript documents an important baseline for assessing future changes at this site, the reporting of the data is quite laborious to follow, and by trying to broaden the scope to assess conditions across the wider region, I think the discussion loses a lot of focus. I am not against the idea of using the other data products to bolster the understanding of the regional context, but I think the authors could make a clearer connection with the main study site. The extensive use of acronyms coupled with quite intricate data reduction methods makes the paper hard to read. My recommendation is that the manuscript needs major revisions to address the substantive points listed below, but I would also encourage the authors to consider ways to improve readability, perhaps by reducing the acronyms and perhaps by focusing the description of the results a bit more to the aspects that they wish to stress during the discussion/conclusions section.

Substantive comments:

1) Study Area section: much of this section reads as an extension of the Introduction, in its detailed description of literature focused on tropical dynamics together with references to previous observational work in the region. However there is no actual description of the local terrain nor mention of other geographical features pertinent to the study. Many readers will not be familiar with Puerto Rico and/or the wider region and so a detailed map or at least a text description documenting the location of the ceilometer and other relevant locations such as the ASOS sites (you give coordinates for the TMCF, but in reality it is not a point. I had to wait for the Methods section to get a brief description of the ceilometer site). If you choose to do a map, you could also indicate the terrain contours in detail and show the proximity to the coast, both of which are very important to the discussion. I would recommend that the current content of this section be worked into the Introduction. 2) Cloud base statistics methods: In Page 3 lines 14-21 there is a thorough description of the method the authors used to generate various statistics for cloud base. While I understand that broken cloud, multiple layers and/or rapidly changing conditions may justify a more detailed algorithm

for identifying cloud base characteristics, I think the current method involving quartiles, tertiles and octiles to produce a set of four cloud base metrics is really confusing. This becomes more confusing when these metrics are then displayed in a histogram type format, because it is hard to tell which of these metrics is most relevant to the ecosystem health. Could the histogram not display the raw 30-second resolution data and that way the frequency counts could be related to a physical diagnostic (i.e. time-in-cloud)? Unless there is an ecosystem relevance to the various statistical quantities, (do they capture/differentiate the intermittency or variability of the cloud within hourly or daily timescales?) the authors should reconsider a more readily interpretable set of metrics. If there are specific reasons, concerning the ecosystem and/or hydrology, for the specific choice of quartiles, tertiles and octiles, then a description of this is certainly warranted. Such a description would be useful for future work, if the same metrics were carried over. 3) CALIPSO: The authors should be careful in their usage of the CALIPSO cloud mask for the purposes they report. Lidar signal attenuates within optically thick clouds and so it is not possible to determine cloud thickness in that case. Winker et al. (2009) report a cloud optical depth of 5 as the threshold below which thickness can be determined and for optically thicker clouds, only the cloud top altitude is possible. Trade cumuli would typically be classified as optically thick using this threshold. On p5 L32, Winker et al. (2009) could be a more appropriate reference than Hunt et al. (2009), since that reference makes no mention of the cloud data products. When using the vertical feature mask, if there is "no signal" data below "cloud" data it is not possible to determine thickness. 4) LCL calculation (described in Appendix A): the authors provide a brief description of calculations, which were done to determine the LCL. The LCL is a parcel property (i.e. given the temperature, pressure and a humidity variable – RH, dew point, mixing ratio. . . - the LCL altitude, LCL pressure and LCL temperature can be uniquely defined). The authors state that surface observations are used, which is an acceptable choice, and they calculate LCL temperature with appropriate citation of the method. However at that point they also have the LCL altitude, by definition. Instead they describe an interpolation of the LCL temperature to determine a corresponding

altitude on a radiosonde sounding. It is not clear to me what that altitude means, but it is not the LCL. This should be addressed before the paper is published.

D.M. Winker, M.A. Vaughan, A. Omar, et al. Overview of the CALIPSO mission and CALIOP data processing algorithms J. Atmos. Ocean. Technol., 26 (11) (2009), pp. 2310-2323

Minor comments:

P1 L30: "Smaller mountains have lower temperatures and steeper adiabatic lapse rates". Please consider rewording this. It is at odds with Line 27-28, and also do you mean pseudo-adiabatic lapse rates? The word steeper is generally confusing when describing lapse rates because of the conventional way of plotting them.

P3 L16 "more diurnal effect of convection" Suggest rewording to make this statement less vague

P3 L24 "originate with strong winds. . ." is this relevant to your site? Later you provide another reference suggesting that winds are 3-5m/s, which is quite light.

P3 L25 suggest "lifting" instead of "forcing"

P4 L3-4 "Raw ceilometer. . ." what you describe is not really raw data, it is a processed product.

P4 L9 How is temporal and spatial variability separated with a ceilometer? This whole sentence is vague, consider rewording.

P4 L13-14 ". . .more complete picture of the climate of the entire atmosphere above the site." Consider replacing "atmosphere" with "troposphere" since you measure <8.2 km. Consider also replacing "climate" with something more specific to the measurement like "cloud patterns" or "cloud variability"

P4 L28-31 Ceiba is only a few km from the site. MSLP is certainly going to be very similar. If this is an example to support the claim that "weather immediately around

the TMCF was homogeneous in pattern..." it is quite weak. Consider removing it. Also the statement "the weather immediately around the TMCF was homogeneous in pattern, but clearly not in magnitude" is vague but also confusing. Later, you go on to show various heterogeneities (associated with the topography and other features) that appears to be in direct contradiction with this statement. Please clarify what you mean and consider using another term instead of "weather".

P4 L34 "climate oscillations" – I think you should be more specific. Also, I think you should provide a bit more clarity on why you removed all the various segments. Was it because you could not clearly define the break between seasons? If so, do you think there is another way of classifying your seasonal groups rather than calendar months?

P7 L29 "within 100 m the LCL" missing "of"?

P8 L3 Suggest stating which stations you are referring to instead of "Caribbean ASOS stations windward of Luquillo" or else, perhaps substitute "windward" for "east"

P8 L3-4 "stable" do you mean that the marine boundary layer is thermodynamically stable?

P8 L5 "remnants of this weather pattern may be carrying on to land..." this is an awkward statement, consider rewording. Also in reference to the previous point about homogeneous "weather" this statement also seems at odds.

P8 L27-29 Please consider rewording this whole sentence. In its present state the meaning is unclear and the wording may need adjusted (e.g. "...evidence that consistently as low, or lower, clouds exist...")

---

## Author Comment (AC2) · 5 May 2017

D. Baumgardner: This manuscript is a revision of a submission that I reviewed prior to its publication as a discussion. I am pleased that the authors acted on some of my recommendations with respect to expanding its scope and supplementing the ceilometer measurements with satellite and radiosonde data. In my opinion the current submission is much improved and I recommend publication. The results are compelling and provide useful information about clouds that impact Tropical Montane Forests. The only recommendation that I have is to add more explanation about the metrics that are used in defining the cloud bases. I would suggest making this part of the Appendix. A representative frequency histogram of cloud bases, such referred in the text, could be
used to mark the various quartiles and tertiles that are being used to define cloud base minimums. In this same appendix it would be useful to further explain to the reader why these particular metrics are optimum, as there are no references to other studies that might have used them. If this is a first time such metrics have been used, then future studies by other investigators would have a cite-able reference if they use the same metrics.

A. Van Beusekom on behalf of the authors: We thank D. Baumgardner for his comments on the earlier version of this manuscript which led to the satellite data being analyzed in the study, and we are pleased he is happy with the expanded scope. With regards to the new comment, we have added a supplement to the manuscript that outlines the workflow of the calculations of metrics on two random days in different seasons. A text section explains why and how we chose the metrics, and how another study could design similar metrics. The histograms of the raw data per day are shown, with the values of the metrics derived from them. The histograms of the hourly metrics used to make the daily metrics are also shown.

---

## Author Response (AR1)

Interactive comment on acp-2016-1166: "Analyzing cloud base at local and regional scales to understand tropical montane cloud forest vulnerability to climate change"

This paper describes results of an extensive cloud and meteorological monitoring campaign in the Caribbean in order to better understand the distribution of low boundary layer marine clouds in time and space. Monitoring was uniquely carried out in mountain regions to gain insights about cloud immersion of tropical montane cloud forests and establish baseline conditions against which future changes may be measured. There are many results that I believe most readers (including me)

10 will find cumbersome to wade through, but that is the nature of such a paper that reports on an extensive field campaign and important general conclusions were still clear. Low clouds are frequent in the year round, with the highest frequency occurring in the dry seasons. Spatial distributions of cloud frequency and base height were quite spatially heterogeneous, however, likely due to topographic effects. Although it is commonly believed that tropical montane cloud forests would be most sensitive to changes in the frequency and elevation of dry-season clouds, authors interpret their detection of a common

15 presence of wet season clouds to indicate that changes to these clouds could also be ecologically impactful. This paper describes critical ground-work and initial results necessary for long-term monitoring of cloud variations and changes in a unique and ecologically important zone. The complicated nature of the results is just the nature of the beast for exploratory studies such as this one that include multiple sites, multiple metrics for measuring mean cloud height, and multiple monitoring tools test for corroboration. I therefore believe this paper is appropriate for publication in ACP.

Specific comments:

P1 L29: It is not explicitly clear what "higher" is relative to.

We changed "higher rainfall" to "more rainfall".

25 P2 L2: I would also imagine fog water input varies greatly throughout a forest depending on they types of surfaces available for collection and micro-climatic wind patterns.

We agree and hope to explore this in future work. We added "this value varies greatly from forest to forest and within the forest."

30 P2 L20-22: It is worth mentioning a couple other relevant studies that have used longterm cloud-base observations to investigate spatiotemporal variations and trends in cloud frequency and cloud-base height. Williams et al. (2015, GRL) used records from 24 airfields in southern California to show that substantial warming-induced increases in cloud base have corresponded to local land-cover change via nighttime urban warming. Richardson et al. (2003, J Climate) analyzed cloud heights at 24 airfields in the Appalachian Mountain region of the eastern US and

35 documented multi-decade (1973-

1999) increases (decreases) in cloud-base height north (south) of 37.5_N. Causes of the changes are not rigorously investigated in that study but the ecological implications are stressed as important. Further, Rastogi et al. (2015, Earth Interactions) presented a relevant study that combined cloud-base observations with satellite and radiosonde observations to estimate how marine-layer stratus clouds intersect with topography.

5   Rastogi, B., A. P. Williams, D. T. Fischer, S. Iacobellis, K. McEachern, L. M. V. Carvalho, C. Jones, S. A. Baguskas, C. J. Still (2015), Spatial and temporal patterns of cloud cover and fog inundation in coastal California: ecological implications, Earth Interactions, In Review.

Richardson, A. D., E. G. Denny, T. G. Siccama, X. Lee (2003), Evidence for a rising cloud ceiling in eastern North America, Journal of Climate, 16(12), 2093-2098, doi:10.1175/1520-0442(2003)016<2093:EFARCC>2.0.CO;2.

10  Williams, A. P., R. E. Schwartz, S. Iacobellis, R. Seager, B. I. Cook, C. J. Still, G. Husak, J. Michaelsen (2015), Urbanization causes increased cloud-base height and decreased fog in coastal southern California, Geophysical Research Letters, 42(5),1527-1536, doi:10.1002/2015GL063266.

We had cited Rastogi et al. 2015 (see pg 17, line 30), in the Methods, because some of the ways they used data sets were
15  similar to ours. We added another citation of Rastogi et al. to the Discussion where we hypothesized the trade wind inversion is limiting the clouds; Rastogi noted that too; thank you for pointing that out. We added a citation of Richardson in the Discussion (pg 21 line 16) discussing possible ecological implications. We added a citation of Williams in the Introduction, saying that future urbanization may affect cloud height (pg 13 line 19).

20      There are many non-traditional abbreviations that I find distracting. For example: ERS, LRS, DS, MSD, TWI. By the half-way point of page 3 I'm fearing that at some point the entire paper may be abbreviated. I recommend unabbreviating some or all of the abbreviations that I list above and I believe doing so will enhance the interpretability
and impact of the paper.

Other reviewers had this comment as well. We removed the seasonal abbreviations but left in the "trade wind inversion" with a TWI abbreviation; it is used in numerous publications and is better known than the rest of the abbreviations that had been used.

30      P4 L2-3: I think there is a word missing from this sentence.

The sentence now reads "Ceilometer data used in this study were the altitudes of the lowest cloud layer at a point above the instrument; the cloud layer base is the bottom of a vertically continuous layer at least 100 m thick with no vertical visibility (defined according to a 5% contrast threshold; http://www.vaisala.com)."

P4 L28-30: It is not indicated what the time-step is for the values considered in this correlation analysis. Daily or annual cycles would cause correlation even if variability that is independent of the cycles is not correlated.

5   We added the word "hourly". The hourly data was used to correlate with clouds on a 5-day and monthly basis.

Interactive comment from D. Baumgardner

Interactive comment on acp-2016-1166: "Analyzing cloud base at local and regional scales to understand tropical montane cloud forest vulnerability to climate change"

This manuscript is a revision of a submission that I reviewed prior to its publication as a discussion. I am pleased that the authors acted on some of my recommendations with respect to expanding its scope and supplementing the ceilometer measurements with satellite and radiosonde data. In my opinion the current submission is much improved and I recommend publication. The results are compelling and provide useful information about clouds that impact Tropical Montane Forests.

The only recommendation that I have is to add more explanation about the metrics that are used in defining the cloud bases. I would suggest making this part of the Appendix. A representative frequency histogram of cloud bases, such referred in the text, could be used to mark the various quartiles and tertiles that are being used to define cloud base minimums. In this same appendix it would be useful to further explain to the reader why these particular metrics are optimum, as there are no

15 references to other studies that might have used them. If this is a first time such metrics have been used, then future studies by other investigators would have a cite-able reference if they use the same metrics.

We have added a supplement that outlines the workflow of the calculations of metrics on two random days in different seasons. A text section explains why and how we chose the metrics, and how another study could design similar metrics. The

20 histograms of the raw data per day are shown, with the values of the metrics derived from them. The histograms of the hourly metrics used to make the daily metrics are also shown.

Anonymous Referee #3

Received and published: March 1, 2017

Interactive comment on acp-2016-1166: "Analyzing cloud base at local and regional scales to understand tropical montane cloud forest vulnerability to climate change"

This paper describes measurements of cloud base altitude using ground-based remote sensing at a site in close proximity to the Luquillo Mountains, Puerto Rico. The authors stress the importance of cloud immersion to the current ecosystem of the tropical montane cloud forest, while also advising the reader that the various mechanisms by which these ecosystems exchange water with the atmosphere are still not fully understood.

10 The authors suggest that rising cloud base altitude as a result of climate change would stress vulnerable species even if rainfall rates in these regions were to remain high. In addition, they hypothesize that the lack of cloud water deposition and increased evapotranspiration, resulting from elevating cloud base, could affect the watershed dynamics in the mountains. The authors also project that their study site may also be vulnerable to changes in the wet season, especially during wet season drought periods.

15 While the manuscript documents an important baseline for assessing future changes at this site, the reporting of the data is quite laborious to follow, and by trying to broaden the scope to assess conditions across the wider region, I think the discussion loses a lot of focus. I am not against the idea of using the other data products to bolster the understanding of the regional context, but I think the authors could make a clearer connection with the main study site. The extensive use of acronyms coupled with quite intricate data reduction methods makes the paper hard to read. My recommendation is that the

20 manuscript needs major revisions to address the substantive points listed below, but I would also encourage the authors to consider ways to improve readability, perhaps by reducing the acronyms and perhaps by focusing the description of the results a bit more to the aspects that they wish to stress during the discussion/conclusions section.

We felt that the regional context was very important to the understanding of precipitation sources for this TMCF (as well as others at similar latitudes). Because the trade winds are a major factor in the climate, changes in climate that affect the forest

25 may be driven more by regional than by local processes. Recognizing the spatial variability, lifting and possible additional cloud formation due to the mountain terrain, we did not assume that the cloud base altitudes measured above the ceilometer were exactly the same as at the forest location. We are currently working on the temporal correlation between the ceilometer observations (from foothills about 7 km upwind of the forest) with simultaneous observations of cloud immersion within the forest. The purpose of the present paper was to develop a method and baseline for long-term measurements and to focus on

30 the regional system of clouds and how it may relate to TMCFs in the trade-wind latitudes. We have made an effort to bring the regional and local aspects of the research together in the discussion, but given the complexity of the subject it will take more than one paper to address the entire set of research questions. To address the readability of this paper, subsection headers were added to the methods and results to discuss each data-type in turn, and clarify the connections between the data-types. We have removed the seasonal abbreviations except in the figures, and added an illustrated supplement to explain

the data reduction (as discussed in comment #2). Numerous revisions and clarifications were added (as recommended by this review and others). We hope these changes make the paper much more readable.

Substantive comments:

1) Study Area section: much of this section reads as an extension of the Introduction, in its detailed description of literature focused on tropical dynamics together with references to previous observational work in the region. However there is no actual description of the local terrain nor mention of other geographical features pertinent to the study. Many readers will not be familiar with Puerto Rico and/or the wider region and so a detailed map or at least a text description documenting the location of the ceilometer and other relevant locations such as the ASOS sites (you give coordinates for the TMCF, but in reality it is not a point. I had to wait for the Methods section to get a brief description of the ceilometer site). If you choose to do a map, you could also indicate the terrain contours in detail and show the proximity to the coast, both of which are very important to the discussion. I would recommend that the current content of this section be worked into the Introduction.

We have added material on the study area location and terrain to the Study Area section. The specific details of the ceilometer location are given in the methods section for the ceilometer, similarly, information on specific details of all the other data types are in the newly added subsections on each data-type. The "study-area" section is meant to provide a broader description of the region and how the data-types fit together. This sentence was added to the end of the study area section "Cloud base height data have been collected in the region by satellites and airport stations in addition to the newer data collected by this study nearer to the TMCF. These data types will be discussed extensively in the next section and a map of locations can be seen in Fig. 4a."

2) Cloud base statistics methods: In Page 3 lines 14-21 there is a thorough description of the method the authors used to generate various statistics for cloud base. While I understand that broken cloud, multiple layers and/or rapidly changing conditions may justify a more detailed algorithm for identifying cloud base characteristics, I think the current method involving quartiles, tertiles and octiles to produce a set of four cloud base metrics is really confusing. This becomes more confusing when these metrics are then displayed in a histogram type format, because it is hard to tell which of these metrics is most relevant to the ecosystem health. Could the histogram not display the raw 30-second resolution data and that way the frequency counts could be related to a physical diagnostic (i.e. time-in-cloud)? Unless there is an ecosystem relevance to the various statistical quantities, (do they capture/differentiate the intermittency or variability of the cloud within hourly or daily timescales?) the authors should reconsider a more readily interpretable set of metrics. If there are specific reasons, concerning the ecosystem and/or hydrology, for the specific choice of quartiles, tertiles and octiles, then a description of this is certainly warranted. Such a description would be useful for future work, if the same metrics were carried over.

We have added a supplement to show how calculation of the frequency-distribution based metrics works out in practice. For low cloud characterization, we wanted to know the cloud base values that occur consistently. We agree that the question could be stated as how often do we see cloud base at a certain altitude range but we concluded that presenting the entire

5 frequency distribution of observations offers the most information about the atmospheric profile to 8 km, and may allow comparison to other forests, as discussed in the section and now extensively in the new supplement. We will address time-in-cloud in a future publication, as explained above, when we have a good understanding of how the ceilometer observations correlate with immersion within the forest. We have rewritten this section extensively to explain the quantiles. For example, before introducing the specific metric quantiles, we say "The specific quantiles used for metrics were chosen such that

10 hourly metric values were between 600-1077 m a majority of the hours in each season and daily metric values were between 600-1077 m a majority of the days in each season. In this way, the metrics can be applied to help quantify the ecosystem characteristic of low-elevation cloud amount needed to sustain the forest throughout the hour, day, and season." Direct frequency histograms by season were made, but did not sufficiently answer the questions of frequency and elevation of low clouds. We hope that by rewriting this section and including the supplement these intentions are clearer.

3) CALIPSO: The authors should be careful in their usage of the CALIPSO cloud mask for the purposes they report. Lidar signal attenuates within optically thick clouds and so it is not possible to determine cloud thickness in that case. Winker et al. (2009) report a cloud optical depth of 5 as the threshold below which thickness can be determined and for optically thicker clouds, only the cloud top altitude is possible. Trade cumuli would typically be

20 classified as optically thick using this threshold. On p5 L32, Winker et al. (2009) could be a more appropriate reference than Hunt et al. (2009), since that reference makes no mention of the cloud data products. When using the vertical feature mask, if there is "no signal" data below "cloud" data it is not possible to determine thickness.
D.M. Winker, M.A. Vaughan, A. Omar, et al. Overview of the CALIPSO mission and CALIOP data processing algorithms J. Atmos. Ocean. Technol., 26 (11) (2009), pp. 2310-2323

Thank you for the suggested reference. The attenuation issue is a very good point—the original calculation removed the attenuated measurements but after more consideration of the systematic error this would introduce we have decided to include all the measurements and use bounds. This is now explained in the methods under subsection 3.4: "Aproximately 10% of the vertical profiles had uncertainty in the cloud base as the lidar was completely attenuated below a cell of cloud

30 (because the cloud was too thick optically; Winker et al., 2009). For these profiles, low cloud base altitude was set as minimally the altitude of land/water surface, and maximally as the lowest cell vertically that cloud is observed, before complete lidar attenuation." We put these ranges into the results subsection (4.4).

4) LCL calculation (described in Appendix A): the authors provide a brief description of calculations, which were

Correct—thank you for pointing this out; we incorrectly used the actual lapse rate to calculate the LCL instead of using the dry adiabatic rate as it should be by definition. We have redone this calculation completely as a mean layer (ML) LCL as the MLLCL was recommended to us as a better representation of the atmospheric profile. We have corrected the description in the Appendix. Subsection 4.3 (results of radiosonde) and figure 4 are redone with the new numbers.

Minor comments:

P1 L30: "Smaller mountains have lower temperatures and steeper adiabatic lapse rates". Please consider rewording this. It is at odds with Line 27-28, and also do you mean pseudo-adiabatic lapse rates? The word steeper is generally confusing when describing lapse rates because of the conventional way of plotting them.

We changed these two sentences to: "Around 500 TMCFs have been identified world-wide on mountains with frequent cloud cover; these can be at higher elevation (on larger mountains) which have lower temperatures, or lower elevation (on smaller mountains) which have more rainfall (Jarvis and Mulligan, 2011). The global set of TMCFs are almost all within 350 km of a coast and topographically exposed to higher-humidity air (Jarvis and Mulligan, 2011). Smaller mountains are likely to have clouds at lower elevations due to slightly higher adiabatic lapse rates (more temperature loss with the same elevation gain) than larger mountains, which undergo greater heating of the land mass (the mass-elevation effect: Foster, 2001; Jarvis and Mulligan, 2011). This effect and the higher humidity near the ocean support TMCFs on small coastal mountains." This statement follows Jarvis and Mulligan who use the term "adiabatic lapse rate" as a general term.

P3 L16 "more diurnal effect of convection" Suggest rewording to make this statement less vague

Changed sentence to: "A pattern of lower clouds (bases and tops) over the ocean and higher clouds over the land with a stronger diurnal effect of convection above land has been observed in the tropics with Cloud-Aerosol Lidar and Infrared Pathfinder Satellite Observation (CALIPSO) (Medeiros et al., 2010)."

P3 L24 "originate with strong winds . . ." is this relevant to your site? Later you provide another reference

Changed to "originate with consistent winds" – which is more appropriate wording as you say. The reference here is from work in the same area where they are talking about the winds in our study region.

P3 L25 suggest "lifting" instead of "forcing"

Changed.

10   P4 L3-4 "Raw ceilometer: : :" what you describe is not really raw data, it is a processed product.

Good point, we've removed all the "raw" words in the paper.

P4 L9 How is temporal and spatial variability separated with a ceilometer? This whole sentence is vague, consider
15   rewording.

We have made our intentions more clear, now saying "we developed metrics to summarize the cloud base altitude data in ways that could express differences and changes temporally (as well as spatially if the same metrics where calculated elsewhere) . . . .."

P4 L13-14 " . . . more complete picture of the climate of the entire atmosphere above the site." Consider replacing "atmosphere" with "troposphere" since you measure <8.2 km.Consider also replacing "climate" with something more specific to the measurement like "cloud patterns" or "cloud variability"

25   Changed to "troposphere" and "cloud patterns"

P4 L28-31 Ceiba is only a few km from the site. MSLP is certainly going to be very similar. If this is an example to support the claim that "weather immediately around the TMCF was homogeneous in pattern . . ." it is quite weak. Consider removing it. Also the statement "the weather immediately around the TMCF was homogeneous in pattern,
30   but clearly not in magnitude" is vague but also confusing. Later, you go on to show various heterogeneities (associated with the topography and other features) that appears to be in direct contradiction with this statement. Please clarify what you mean and consider using another term instead of "weather".

The stations we were referring to are the ones in the forest, albeit at different locations, and not Ceiba. We've reworded these

sentences to avoid misunderstanding: "were collected at several weather stations around the TMCF within an 8 km$^2$ area of the forest for differing periods of record (PORs) (Table 1). High hourly correlations were observed between these parameters across the forest stations for periods of overlap, giving confidence that patterns between the set of weather variables at each station in the immediate vicinity of the ceilometer measurements were homogeneous, although of differing magnitude at the different stations (Van Beusekom et al., 2015). Thus we used the weather data from the one station with the most complete and longest POR (Bisley at 361 m; Table 1). Mean sea level pressure (MSLP) was only collected near the TMCF at 100 m. Its relatively short record was highly correlated with the MSLP at the ASOS station TJNR, Ceiba (correlation coefficient $\rho =$ 0.98), so the record was filled with data from Ceiba."

P4 L34 "climate oscillations" – I think you should be more specific. Also, I think you should provide a bit more clarity on why you removed all the various segments. Was it because you could not clearly define the break between seasons? If so, do you think there is another way of classifying your seasonal groups rather than calendar months?

Changed to "However, interannual variability in the timing of seasons from climate oscillations (North Atlantic, Pacifica Decadal, and El Niño-Southern Oscillations) was observed in averages made over the shorter record (Gouirand et al., 2012)." We've changed the next sentence to "Because we have a short period of record for the ceilometer data (~3 years) and we focused on average seasonal behavior in this initial study, we chose to omit the unrepresentative time periods noted above and only used the time periods representative of the longer-term seasonal average (~17 years). As the ceilometer record gets longer, we will be better able to investigate effects of the large-scale climate oscillations on clouds at the site." We could have done a more sophisticated analysis to break the seasons—but past work has shown efforts to be complicated and we feel that with three years of data that analysis would just add confusion. (See Van Beusekom, A.E., González, G., Rivera, M.M., 2015. Short-Term Precipitation and Temperature Trends along an Elevation Gradient in Northeastern Puerto Rico. Earth Interact. 19, 1–33.).

P7 L29 "within 100 m the LCL" missing "of"?

Yes, changed.

P8 L3 Suggest stating which stations you are referring to instead of "Caribbean ASOS stations windward of Luquillo" or else, perhaps substitute "windward" for "east"

Changed to "Caribbean ASOS stations east (windward) of Puerto Rico"

Changed to "thermodynamically stable marine boundary layer TWI"

We changed it to "remnants of this weather pattern may be persisting on to the land in eastern Puerto Rico".
This statement shouldn't be at odds with the previous statement about the weather immediately around the forest—hopefully
10 with the clarifications in the earlier statement this now appears un-contradictory.

[revised manuscript text omitted]

Ashley Van Beusekom 5/5/2017 3:52 PM

Ashley Van Beusekom 5/5/2017 3:52 PM

Ashley Van Beusekom 5/5/2017 3:52 PM

Ashley Van Beusekom 5/5/2017 3:53 PM

| Page 15: [1] Deleted | Ashley Van Beusekom | 5/5/17 2:28 PM |
| --- | --- | --- |

| Page 15: [1] Deleted | Ashley Van Beusekom | 5/5/17 2:28 PM |
| --- | --- | --- |

| Page 15: [1] Deleted | Ashley Van Beusekom | 5/5/17 2:28 PM |
| --- | --- | --- |

| Page 15: [2] Deleted | Ashley Van Beusekom | 5/5/17 3:30 PM |
| --- | --- | --- |

| Page 15: [2] Deleted | Ashley Van Beusekom | 5/5/17 3:30 PM |
| --- | --- | --- |

| Page 15: [2] Deleted | Ashley Van Beusekom | 5/5/17 3:30 PM |
| --- | --- | --- |

| Page 15: [3] Deleted | Ashley Van Beusekom | 2/14/17 8:49 AM |
| --- | --- | --- |

of

| Page 15: [3] Deleted | Ashley Van Beusekom | 2/14/17 8:49 AM |
| --- | --- | --- |

of

| Page 15: [4] Deleted | Ashley Van Beusekom | 2/14/17 2:38 PM |
| --- | --- | --- |

defined as the cloud base (

| Page 15: [4] Deleted | Ashley Van Beusekom | 2/14/17 2:38 PM |
| --- | --- | --- |

defined as the cloud base (

| Page 15: [4] Deleted | Ashley Van Beusekom | 2/14/17 2:38 PM |
| --- | --- | --- |

defined as the cloud base (

| Page 15: [4] Deleted | Ashley Van Beusekom | 2/14/17 2:38 PM |
| --- | --- | --- |

defined as the cloud base (

| Page 15: [5] Deleted | Ashley Van Beusekom | 3/27/17 7:53 PM |
| --- | --- | --- |

s

| Page 15: [5] Deleted | Ashley Van Beusekom | 3/27/17 7:53 PM |
| --- | --- | --- |

s

| Page 15: [5] Deleted | Ashley Van Beusekom | 3/27/17 7:53 PM |
| --- | --- | --- |

s

| Page 15: [6] Deleted | M Scholl | 4/19/17 11:07 AM |
| --- | --- | --- |

all day

| Page 15: [6] Deleted | M Scholl | 4/19/17 11:07 AM |
|---|---|---|

all day

| Page 15: [7] Deleted | Ashley Van Beusekom | 3/24/17 10:30 AM |
|---|---|---|

climate

| Page 15: [7] Deleted | Ashley Van Beusekom | 3/24/17 10:30 AM |
|---|---|---|

climate

| Page 15: [8] Deleted | M Scholl | 4/19/17 11:13 AM |
|---|---|---|

calculated in

| Page 15: [8] Deleted | M Scholl | 4/19/17 11:13 AM |
|---|---|---|

calculated in

| Page 15: [8] Deleted | M Scholl | 4/19/17 11:13 AM |
|---|---|---|

calculated in

| Page 15: [8] Deleted | M Scholl | 4/19/17 11:13 AM |
|---|---|---|

calculated in

| Page 15: [8] Deleted | M Scholl | 4/19/17 11:13 AM |
|---|---|---|

calculated in

| Page 15: [9] Deleted | Ashley Van Beusekom | 2/17/17 8:53 PM |
|---|---|---|

Hourly

| Page 15: [9] Deleted | Ashley Van Beusekom | 2/17/17 8:53 PM |
|---|---|---|

Hourly

| Page 15: [9] Deleted | Ashley Van Beusekom | 2/17/17 8:53 PM |
|---|---|---|

Hourly

| Page 16: [10] Deleted | Ashley Van Beusekom | 5/5/17 3:08 PM |
|---|---|---|

Luquillo

| Page 16: [10] Deleted | Ashley Van Beusekom | 5/5/17 3:08 PM |
|---|---|---|

Luquillo

| Page 16: [11] Deleted | M Scholl | 2/10/17 2:58 PM |
|---|---|---|

immediately

| Page 16: [11] Deleted | M Scholl | 2/10/17 2:58 PM |
|---|---|---|

immediately

| Page 16: [12] Deleted | M Scholl | 4/19/17 11:28 AM |
|---|---|---|

variables at

| Page 16: [12] Deleted | M Scholl | 4/19/17 11:28 AM |
|---|---|---|

variables at

| Page 16: [13] Deleted | Ashley Van Beusekom | 3/24/17 10:57 AM |
|---|---|---|

.

| Page 16: [14] Deleted | Ashley Van Beusekom | 5/5/17 2:32 PM |
|---|---|---|

Luquillo

| Page 16: [15] Deleted | M Scholl | 4/19/17 11:36 AM |
|---|---|---|

are

| Page 18: [16] Deleted | Ashley Van Beusekom | 2/16/17 3:03 PM |
|---|---|---|

.

All correlations between variables were computed as Pearson product-moment correlations with coefficient $\rho$ measuring linear correlation. Strengths of correlations were based on the relationships between weather parameters and clouds found in this study and another Caribbean cloud and weather study (Brueck et al., 2014); correlation was assumed fair if $0.3 < |\rho| < 0.4$, good if $0.4 < |\rho| < 0.6$, and very good if $|\rho| > 0.6$. The regional means and correlations were also computed over the shorter LQ POR with exclusion of the non-representative months. These were not substantially different than the 2000-2016 results presented here (in Fig. 4, and Fig. 5, Table 3), but the mathematical strength of the resulting means and significance of the correlations necessarily decreased with less data.

| Page 19: [17] Deleted | M Scholl | 4/19/17 12:37 PM |
|---|---|---|

drought

| Page 19: [18] Deleted | Ashley Van Beusekom | 5/5/17 2:54 PM |
|---|---|---|

Luquillo,

| Page 19: [18] Deleted | Ashley Van Beusekom | 5/5/17 2:54 PM |
|---|---|---|

Luquillo,

| Page 19: [19] Deleted | Ashley Van Beusekom | 3/23/17 12:35 PM |
|---|---|---|

ERS

| Page 19: [20] Deleted | Ashley Van Beusekom | 3/23/17 12:36 PM |
|---|---|---|

MSD

| Page 19: [21] Deleted | Ashley Van Beusekom | 4/20/17 10:41 AM |
|---|---|---|

 Time?

| Page 19: [21] Deleted | Ashley Van Beusekom | 4/20/17 10:41 AM |
|---|---|---|

Time?

| Page 19: [22] Deleted | Ashley Van Beusekom | 2/14/17 9:30 AM |
|---|---|---|

In the northern Carbbean region, Regionally, the mean low cloud base from the mean-hours was lower during the MSD than during the other seasons at all of the stations, indicating seasonal variation in mean low cloud base altitude (Fig. 4c). However, the longer winter dry season, DS, had a higher low cloud base than the wet seasons except at Luquillo (note that at Luquillo in Fig. 4b, less frequent DS high clouds (see Fig. 1) lowers the mean-hour cloud base). Cloud cover percentage was highest in the MSD relative to the other seasons at all the stations to the east (windward) and including Luquillo, and DS cloud cover was similar to or higher than the wet seasons (Fig. 4d). The low and minimum cloud bases for stations near to Luquillo correlated much better with cloud cover than at the stations as a whole, with very good negative correlation at Luquillo itself (Table 2, S3).

In contrast to Luquillo (Fig. 1b) the regional pattern of RH for 2000-2016 was as expected, with lower RH during the dry seasons than the wet seasons (Fig. 5) and the low and minimum cloud bases all had fair correlation with the RH measurements (Table 3). Regionally, no metrics of daily clouds correlated with MSLP except at Luquillo, where the low and maximum cloud bases had good and very good correlation with MSLP, respectively (Table 3).

| Page 19: [23] Deleted | M Scholl | 2/13/17 10:36 AM |
|---|---|---|

data

| Page 19: [24] Deleted | Ashley Van Beusekom | 5/5/17 1:44 PM |
|---|---|---|

,

**Page 20: [25] Deleted**            **M Scholl**            **2/13/17 10:35 AM**

oceanic

**Page 20: [26] Deleted**            **M Scholl**            **2/13/17 10:35 AM**

land with high topography increased the likelihood of clouds in a layer from land surface elevation up to 2 km, approximately the altitude of the TWI which under most conditions caps the trade-wind cumuli (Fig. 6)

**Page 20: [27] Deleted**            **Ashley Van Beusekom**            **3/13/17 8:28 PM**

845

**Page 20: [28] Deleted**            **Ashley Van Beusekom**            **3/13/17 8:28 PM**

increase in

increase in

| Page 20: [28] Deleted | Ashley Van Beusekom | 3/13/17 8:28 PM |

increase in

| Page 20: [29] Deleted | Ashley Van Beusekom | 2/14/17 9:17 AM |

t was cloudier over land and

| Page 20: [29] Deleted | Ashley Van Beusekom | 2/14/17 9:17 AM |

t was cloudier over land and

| Page 20: [29] Deleted | Ashley Van Beusekom | 2/14/17 9:17 AM |

t was cloudier over land and

| Page 20: [30] Deleted | Ashley Van Beusekom | 3/13/17 8:29 PM |

358

| Page 20: [30] Deleted | Ashley Van Beusekom | 3/13/17 8:29 PM |

358

| Page 20: [30] Deleted | Ashley Van Beusekom | 3/13/17 8:29 PM |

358

| Page 20: [31] Deleted | M Scholl | 4/19/17 1:27 PM |

by

| Page 20: [31] Deleted | M Scholl | 4/19/17 1:27 PM |

by

| Page 20: [32] Deleted | Ashley Van Beusekom | 2/17/17 8:58 PM |

our ceilometer

| Page 20: [32] Deleted | Ashley Van Beusekom | 2/17/17 8:58 PM |

our ceilometer

| Page 20: [32] Deleted | Ashley Van Beusekom | 2/17/17 8:58 PM |

our ceilometer

| Page 20: [33] Deleted | M Scholl | 4/19/17 1:29 PM |

this

| Page 20: [33] Deleted | M Scholl | 4/19/17 1:29 PM |

this

| Page 20: [33] Deleted | M Scholl | 4/19/17 1:29 PM |

this

| Page 20: [34] Deleted | M Scholl | 4/19/17 3:26 PM |

increased

**Page 20: [34] Deleted**          **M Scholl**          **4/19/17 3:26 PM**
increased

**Page 20: [34] Deleted**          **M Scholl**          **4/19/17 3:26 PM**
increased

**Page 21: [35] Deleted**          **M Scholl**          **4/19/17 4:08 PM**
The Caribbean region is projected to undergo drying in the future, with various mechanisms that will suppress deep convection and frequent high rainfall (Karmalkar et al., 2013). If trade-wind clouds were thinner and shallow convection was weak during those periods, drought effects on the TMCF could be significant.